

# Sediment and carbon accumulation vary among vegetation assemblages in a coastal saltmarsh

Jeffrey J. Kelleway[1,2], Neil Saintilan[2], Peter I. Macreadie[1,3], Jeffrey A. Baldock[4], Peter J. Ralph[1]

[1]Plant Functional Biology and Climate Change Cluster, School of Life Sciences, University of Technology Sydney, Ultimo, NSW 2007, Australia
[2]Department of Environmental Sciences, Macquarie University, Sydney, NSW 2109, Australia
[3]School of Life and Environmental Sciences, Centre for Integrative Ecology, Deakin University, Victoria 3216, Australia
[4]CSIRO Land and Water/Sustainable Agriculture Flagship, Glen Osmond, SA 5064, Australia

*Correspondence to*: J. J. Kelleway (jeffrey.kelleway@mq.edu.au)

**Abstract.** Coastal saltmarshes are dynamic, intertidal ecosystems which are increasingly being recognised for their contributions to ecosystem services, including carbon (C) accumulation and storage. The survival of saltmarshes and their capacity to store C under rising sea levels, however, is partially reliant upon surface sedimentation rates and influenced by a combination of physical and biological factors. In this study, we use several complementary methods to assess short-term (days) deposition and medium-term (months) accretion dynamics within three saltmarsh vegetation types common throughout southeast (SE) Australia.

We found that surface accretion varies among vegetation assemblages, with medium-term (19 month) bulk accretion rates in the upper marsh rush (*Juncus*) assemblage (1.74 ± 0.13 mm y$^{-1}$) consistently in excess of estimated local sea level rise (1.15 mm y$^{-1}$). Accretion was lower and less consistent in both the succulent (*Sarcocornia*) (0.78 ± 0.18 mm y$^{-1}$) and grass (*Sporobolus*) (0.88 ± 0.22 mm y$^{-1}$) assemblages located lower in the tidal frame. Short-term (6 d) experiments showed deposition within *Juncus* plots to be dominated by autochthonous organic inputs with C deposition rates ranging from 0.41 ± 0.15 g C cm$^{-2}$ y$^{-1}$ (neap tidal period) to 0.87 ± 0.16 g C cm$^{-2}$ y$^{-1}$ (spring tidal period), while minerogenic inputs and lower C deposition dominated *Sarcocornia* (0.03 ± 0.01 to 0.23 ± 0.03 g C cm$^{-2}$ y$^{-1}$) and *Sporobolus* (0.06 ± 0.01 to 0.15 ± 0.03 g C cm$^{-2}$ y$^{-1}$) assemblages.

Elemental (C:N), isotopic (δ$^{13}$C), mid infrared (MIR) and $^{13}$C NMR analyses revealed little difference in either the source or character of materials being deposited among neap versus spring tidal periods. Instead, these analyses point to substantial redistribution of materials within the *Sarcocornia* and *Sporobolus* assemblages, compared to high retention and preservation of organic inputs in the *Juncus* assemblage. By combining medium-term accretion quantification with short-term deposition measurements and chemical analyses we have gained novel insights into biophysical processes responsible for regional differences in surface dynamics among key saltmarsh vegetation assemblages. Our results suggest that unless belowground processes (e.g. root production) make substantial contributions to surface elevation gain, then *Sarcocornia* and *Sporobolus* assemblages may be particularly susceptible to changes in sea level, with implications for the future structure and function of these saltmarsh areas.



## 1 Introduction

### 1.1 Coastal wetlands

Coastal saltmarshes are dynamic ecosystems, vegetated by herbs, grasses and rushes that are found in a range of sedimentary settings along low-energy coastlines. Globally, vegetation type and floristic assemblage have been used to classify broad types of saltmarsh (Adam, 1990; Adam, 2002). At the local scale, vegetation zonation is one of the most striking ecological features of many saltmarshes, reflecting the elevation requirements of a small number of dominant species, although mosaics of species within a zone are also common (Adam, 2002; Hickey and Bruce, 2010). Whilst the biodiversity values and exceptional productivity of coastal wetlands have been long recognised, increasing attention is now being focused upon ecosystem services such as carbon (C) accumulation and storage (Chmura et al., 2003; Duarte et al., 2013), and the response of these coastal ecosystems to changes in climate (Kirwan and Mudd, 2012) and sea level (Rogers et al., 2013).

Surface elevation and sedimentation dynamics are central to both coastal wetland survival under rising sea level (Baustian et al., 2012; Kirwan and Megonigal, 2013; Kirwin et al., 2016) and to the delivery and storage of organic matter (OM) (Duarte et al., 2013; Lovelock et al., 2013). Saltmarsh sediments may be minerogenic (dominated by mineral inputs) or organogenic (dominated by biomass and litter production and/or allochthonous OM inputs), although most sediments comprise both mineral and organic fractions (Adam, 2002; Baustian et al., 2012). Consequently, sediment properties and surface dynamics may be influenced by both physicochemical and biological factors. Physical drivers of accretion (the vertical accumulation of sediment) in intertidal wetlands include the suspended sediment supply of inundating waters (Zhou et al., 2007); as well as the tidal range of a site and position within the tidal range (Ouyang and Lee, 2014; Saintilan et al., 2013; van Proosdij et al., 2006). High tides may play an important role in importing sediment into saltmarshes (Rosencranz et al., 2015), while low-tide rainfall may act to redistribute or export materials, including particulate organic carbon (Chen et al., 2015).

Numerous studies have investigated the interactions between vegetation and marsh surface dynamics, although the majority of these studies have focussed on the genus *Spartina* (e.g. Baustian et al., 2012; Mudd et al., 2010; Mudd et al., 2009; Nyman et al., 2006). Broadly, these studies have shown that the presence of vegetation may have a significant positive influence on surface accretion through: 1) accumulating organic matter; and 2) helping to trap mineral sediments (Morris et al., 2002; Mudd et al., 2010; Nyman et al., 2006). Comparative studies of the effect of vegetation composition in the intertidal zone, however, vary from no difference in accretion among different vegetation species in the wetland (e.g. Culberson et al., 2004) to sizeable differences among mangroves and different saltmarsh species (e.g. Saintilan et al., 2013).

### 1.2 C storage

Globally, the rate of sediment carbon accumulation is extremely high in saltmarshes, relative to terrestrial and most coastal ecosystems, with a mean ± SE accumulation rate of $0.024 ± 0.003$ g C $cm^{-2}$ $y^{-1}$ (Ouyang and Lee, 2014). While much of this C is produced belowground by roots and rhizomes, contributions from aboveground sources may be significant (Boschker et al., 1999; Zhou et al., 2006). Sources of aboveground C may include both autochthonous (produced within the community) and allochthonous (deposited from outside the community) OM, although their relative contributions may vary within and among saltmarsh settings (Kelleway et al., 2016a). Regardless of OM source, the capacity of coastal wetlands to store carbon in the long-term remains dependent upon the balance between OM inputs and their decay (Kirwan et al., 2013). While there is considerable debate as



to which factors most influence the long-term retention of C in soil pools, litter quality has long been identified as a key driver of decay rates (Cleveland et al., 2014; Enríquez et al., 1993; Josselyn and Mathieson, 1980; Kristensen, 1994) and is of particular relevance to C stock accumulation on the sediment surface.

### 1.3 Measuring surface deposition and accretion

A variety of methods have been developed for measuring and monitoring surface dynamics in tidal wetlands (for reviews see Nolte et al., 2013; Thomas and Ridd, 2004). These include techniques relevant to short-term *deposition* events (of sediments and plant litter) through to medium- and long-term measures of *accretion* or *accumulation* (the net effect of multiple deposition and removal events) as well as *surface-elevation change*. Methods also vary in their effectiveness of trapping and retaining different materials, meaning a combination of

techniques may be required to identify the different physical and biotic influences on deposition and accretion (Nolte et al., 2013). In this study, we use several methods to assess short-term (days) deposition and medium-term (months) accretion dynamics within three saltmarsh vegetation assemblages common throughout southeast (SE) Australia. We hypothesise that: 1) deposition and accretion will vary among assemblages, in accordance with differences in vegetation structure and location within the saltmarsh; and 2) the source and character of material

deposited will vary temporally according to tidal inundation patterns. This study also presented an opportunity to compare wetland sedimentation methods. Together, we expect this information will improve our understanding of how materials (including C) accumulate in coastal wetlands and how these ecosystems might respond under rising sea level.

## 2 Methods

### 2.1 Study setting

Towra Point Nature Reserve is located within the oceanic embayment Botany Bay, approximately 16 km south of central Sydney, Australia's largest city. The intertidal estuarine wetland complex at this site is the largest remaining within the Sydney region and is listed as a Ramsar Wetland of International Importance. Within the site, a large saltmarsh area adjacent to Weeney Bay was chosen as a study site (Fig. 1) as this area exhibits

vegetation zonation typical of SE Australian saltmarshes. The lower saltmarsh is bordered by the mangrove *Avicennia marina*, beyond which seagrass meadows (including *Posidonia australis*) occur within the subtidal zone. In some areas the upslope limit of saltmarsh extends into small patches of the supratidal trees *Casuarina glauca* and *Melaleuca ericifolia,* but for the most part is bordered by a levee which was constructed between 1947 and 1951. Previous investigation has revealed vegetation zonation across the site coinciding with elevation

measurements and tidal extent modelling (Hickey and Bruce, 2010).

Saltmarsh within this site comprises two broad vegetation communities – an association of the perennial succulent *Sarcocornia quinqueflora* (C3 photosynthetic pathway) and the perennial grass *Sporobolus virginicus* (C4 photosynthetic pathway)*,* mostly intermixed across the lower and middle marsh. The upper marsh assemblage is dominated by the rush *Juncus kraussii* (C3)*,* with *S. virginicus* (C4) ubiquitous as a sub-dominant lower stratum

across this assemblage.

Fifteen plots were selected for study on the basis of saltmarsh vegetation zonation – five plots randomly chosen within the *Juncus*-dominated assemblage, and 10 plots strategically selected within the *Sarcocornia-Sporobolus*



association (five plots vegetated exclusively by *Sarcocornia,* and five vegetated exclusively by *Sporobolus*). Hereafter, these three assemblages are referred to by genus (*Sarcocornia; Sporobolus; Juncus*), while reference to the plant species themselves involves the species name (*S. quinqueflora; S. virginicus; J. kraussii*).

Data previously collected within the study region showed a substantial difference in aboveground biomass of the rush assemblage (Juncus mean = 1116 g m$^{-2}$), compared to that of the non-rush assemblages (*Sarcocornia* mean = 320 g m$^{-2}$; *Sporobolus* mean = 350 g m$^{-2}$) - importantly, there do not appear to be distinct seasonal patterns of biomass stock for any of these species (Clarke and Jacoby, 1994). Both *Sarcocornia* and *Sporobolus* are perennial species, while *J. kraussii* culms undergo initiation and senescence throughout the year, but with peak culm initiation before and after summer flowering and fruiting (Clarke and Jacoby, 1994). Belowground biomass data are rare, though on the basis of data presented by Clarke and Jacoby (1994) we have calculated a mean aboveground : belowground biomass ratio of 1.5 for *Juncus*. No belowground data have been reported for either *Sarcocornia* and *Sporobolus*. More broadly, the saltmarshes of southeast (SE) Australia have been classified within the temperate group of saltmarshes which also includes those of Europe, the Pacific coast of North America, Japan and South Africa (Adam, 1990). These are distinct from the well-studied *Spartina*-dominated marshes of North America's Atlantic coast.

Tides along the New South Wales coast are semidiurnal (two flood and two ebb periods each lunar day) with a maximum spring tidal range of 2.0 m (Roy et al., 2001). Astronomical (i.e. predicted) maxima occur during the new moon in summer and during the full moon in winter (spring tides). Tidal inundation to and recession from the study area occurs via Weeney Bay, with the causeway acting as a barrier to surface water exchange with the western section of the Nature Reserve and Woolooware Bay. The linear rate of sea level rise in Botany Bay since local records commenced in 1981 is 1.15 mm y$^{-1}$ (Kelleway et al., 2016b).

### 2.2 Elevation measurement

Within each plot, elevation was measured using a modified version of the tidal inundation method described by English et al. (1994), whereby three vertical rods marked with water-soluble dye were inserted into the ground immediately prior to a summer spring tide (23/01/2015; measured tidal height of 1.897 m above lowest astronomical tide (LAT) datum at nearest tidal gauge). Depth of inundation above the saltmarsh surface was measured immediately after the tide receded and subtracted from the predicted tide height to obtain an estimate of surface elevation. Care was taken during the measurement procedure and in the selection of a calm day (to minimise wind and waves effects) to minimise discrepancies between measurements at different plots. Comparison of three replicate rods revealed a standard error of the mean < 1.3 cm for each plot.

### 2.3 Feldspar marker horizons

The feldspar marker horizon (MH) technique (Cahoon and Turner, 1989) has been proposed as a suitable method to investigate the effects of aboveground vegetation structure on the accretion (vertical accumulation) of material on the marsh surface over the medium-term (Nolte et al., 2013). A total of 45 feldspar MHs were installed across the study site on 23 January 2014, comprising three replicates in each of the 15 study plots. Accretion was determined at later dates as the height difference between the marsh surface and the feldspar (i.e., the material accumulated above the MH), and was recorded as the mean of three replicate measurements from within the marker horizon at each sampling event. Measurements were taken 11, 13, 15, 17 and 19 months after installation.





During the later sampling events many MHs in *Sarcocornia* and *Sporobolus* plots became increasingly difficult to discern within the sediment, probably due to bioturbation and mixing of sediments (Cahoon and Turner, 1989; Krauss et al., 2003). Consequently, monitoring of all plots was terminated after 19 months.

**2.4 Sedimentation traps**

Two complementary types of sedimentation trap were installed concurrently for the purpose of quantifying short-term deposition of materials among the three vegetation assemblages. First, pre-weighed, 50 mL centrifuge vials (30 mm mouth diameter; 115 mm depth) were placed into the ground, so that the 'lip' of each tube was 10 mm above the ground surface. This vial method has a bias towards the collection of non-buoyant materials washing over the mouth of the tube (i.e. mineral matter) and a bias against collection of coarse and/or buoyant materials, including large fragments of plant litter. Second, a modified version of the filter paper method described by Reed (1989) and Adame et al. (2010) was used to quantify 'passive' sedimentation and litter accretion on the saltmarsh surface. Pre-weighed 90 mm hydrophilic nylon filters (pore size 0.45μm) were placed over 90 mm upturned plastic Petri dishes, and attached to the sediment by two small staples, so that the nylon filter lay level with the sediment surface. The resolution of this method, using a 90 mm filter has been calculated as 0.0015 mg cm$^{-2}$ (Thomas and Ridd, 2004).

Three replicates of each short-term trap were installed at the centre of each of the study areas described above during the summer of 2014/15. Traps were deployed for 6 d (12 high tides) periods on four instances, on the basis of tide chart predictions. Two neap ('December neap' and 'January neap') periods were selected to reflect periods when high tides were at their lowest. While these neap periods were intended to measure periods without any inundation, higher than predicted tides occurred in both neap periods. Although unconfirmed, inundation of some plots within lower elevation zones of the study area were expected to have occurred at least once during the December neap (up to 80% of *Sarcocornia* plots and 100% of *Sporobolus* plots) and/or the January neap (up to 60% of *Sarcocornia* plots only) (Table S1). Two other periods ('December spring' and 'January spring') were selected as maximum saltmarsh inundation events with between five and ten high tides inundating each plot in each period (Table S1). Although unintended, the fact that a small number of inundations were likely captured during neap tides more accurately reflects the differences in tidal behaviour that naturally occurs among the three vegetation assemblages (i.e. lower elevation assemblages are subject to a greater number of high tides throughout the year than higher elevation assemblages). Consequently, all results from short-term measures were considered in the context of these varied inundation patterns.

Great care was taken not to disturb sediments or litter collected on, or surrounding the removable traps during their installation and collection. Filters with visible crab-excavated sediment (n = 23/180) or physically upturned during inundation (n = 3 January spring inundation only) were excluded from analysis, although all plant (autochthonous and allochthonous) materials were retained for analysis as we considered these to be largely unaffected by crab excavation.

In the laboratory, vials were centrifuged, the supernatant decanted and vial placed in an oven for drying. All samples and vessels (filters and centrifuge vials) were dried at 60ºC until constant weight was achieved (≤ 72 h) and subtracted from initial vessel mass to obtain the dry weight of material collected. In addition, all identifiable litter was removed from each filter, identified to the species level and weighed. Litter samples of the main saltmarsh species encountered (*S. quinqueflora, S. virginicus* and *J. kraussii*), wrack of the seagrass *Posidonia*



*australis* and macroalga *Hormosira banksii*, fresh leaves of the mangrove *Avicennia marina,* as well as composite samples of all residual sediment (mineral component and unidentified organic matter; referred hereafter as 'residue') from filters were also prepared for chemical analyses.

### 2.5 Elemental and isotopic analysis

Dried aboveground plant biomass, litter and residues were homogenised and ground into a fine powder using a ball mill. The 'Champagne test' (Jaschinski et al., 2008) was used to determine that no residue samples contained inorganic C. Consequently, acidification of samples was deemed unnecessary. Organic %C, %N, and $\delta^{13}$C were measured for all samples using an Isotope Ratio Mass Spectrometry – Elemental Analyzer (Thermo DeltaV) at University of Hawaii (HILO).

### 2.6 MIR

Diffuse reflectance mid-infrared (MIR) spectroscopy was used to assess the composition of biomass, litter and residue samples. MIR spectroscopy characterises the bulk composition and is therefore inclusive of both mineral and organic components. Spectra were acquired using a Nicolet 6700 FTIR spectrometer (Thermo Fisher Scientific Inc., Waltham, MA, USA) following the specifications and procedures outlined by Baldock et al.

(2013a). Spectra were acquired over 8000–400 cm$^{-1}$ with a resolution of 8 cm$^{-1}$, but were truncated to 6000–600 cm$^{-1}$. Spectra were baseline-corrected using a baseline-offset transformation and then mean-centred using the Unscrambler 10.2 software (CAMO Software AS, Oslo, Norway) before conducting principal component analysis (PCA).

### 2.7 $^{13}$C NMR

Solid-state $^{13}$C nuclear magnetic resonance (NMR) spectroscopy was used to quantify the contribution of C functional groups to live plant biomass, litter and residue samples. Residue samples were treated with 2% hydrofluoric acid (HF) according to the method of Skjemstad et al. (1994) to remove paramagnetic materials and concentrate organic C for $^{13}$C NMR analyses. Cross-polarization $^{13}$C NMR spectra were acquired using a 200 MHz Avance spectrometer (Bruker Corporation, Billerica, MA, USA) following the instrument specifications,

experimental procedures and spectral processing outlined by Baldock et al. (2013b). $^{13}$C NMR data are presented as the proportion of integral area under each of eight chemical shift regions corresponding to main types of organic functional groupings found in natural organic materials: Alkyl C (0-45 ppm), *N*-Alkyl/Methoxyl (45-60 ppm), *O*-Alkyl (60-95 ppm), Di-*O*-Alkyl (95-110 ppm), Aryl (110-145 ppm), *O*-Aryl (145-165 ppm), Amide/Carboxyl (165-190 ppm) and Ketone (190-215 ppm) (Baldock and Smernik, 2002).

### 2.8 Statistical analyses

Separate simple linear regression analyses were conducted using all feldspar MH measurements for each of the three vegetation assemblages for the purpose of obtaining accumulation rates over 19 months and to assess the strength of linear fits for these data. Bulk short-term deposition variables (bulk material collected in vials; bulk material collected on filters) were log-transformed to achieve normality and analysed with separate linear mixed

models, to test main and interactive effects of vegetation assemblage (*Sarcocornia*, *Sporobolus*, *Juncus*) and tidal event (repeated measures: December neap, December spring, January neap, January spring). Elevation was





included as a covariate for each of these analyses. Covariance structure was selected for each model through comparison of Akaike's Information Criterion (AIC) of four covariance structures (unstructured, compound symmetry, diagonal, scaled identity). Where main effects presented significance differences ($P < 0.05$), post hoc tests (with conservative Bonferroni adjustment) were used to determine difference among levels of vegetation and tidal event factors. Statistical analyses were performed using SPSS v19 (IBM, USA), Origin Pro 2015 (OriginLab, USA) and PRIMER v6 (PRIMER-E, UK).

### 2.8.1 Isotope mixing model

A two-source, single isotope mixing model (Phillips, 2012) was used to estimate the proportion of C3 ($f_1$ in equation (1)) and C4 ($f_2$ in equation (2)) plants to the unidentified organic residue:

$$f_1 = \frac{\delta^{13}C_{residue} - \delta^{13}C_{C4}}{\delta^{13}C_{C3} - \delta^{13}C_{C4}}$$
(1)

$$f_2 = 1 - f_1$$
(2)

where $C_{residue}$ denotes the residue organic C, $C_{C3}$ denotes the $\delta^{13}C$ of the relevant C3 plants (*S. quinqueflora* litter for *Sarcocornia-Sporobolus* association residues or *J. kraussii* litter for *Juncus* assemblage residues) and $C_{C4}$ denotes $\delta^{13}C$ of litter of the C4 species (*S. virginicus*).

### 2.8.2 MIR analysis

Principal components analysis (PCA) was performed using the transformed MIR spectra to: 1) identify differences in composition among samples due to sample type and vegetation assemblage; and 2) define the MIR spectral components most important to differentiating the samples. Loadings were plotted for the first two principal components to assist in the latter and to guide interpretation of differences in composition among samples.

## 3 Results

### 3.1 Feldspar MHs

Positive and consistent accretion was measured among *Juncus* plots throughout the entire 19 months, reflected in the moderate-strong linear fit ($R^2 = 0.68$; $P <0.001$) and a mean accumulation rate with relatively low variance ($1.74 \pm 0.13$ mm y$^{-1}$). In contrast, accumulation above the feldspar MHs was more varied and slower overall in the *Sarcocornia* ($R^2 = 0.16$; $P <0.001$; $0.76 \pm 0.18$ mm y$^{-1}$) and *Sporobolus* plots ($R^2 = 0.14$; $P <0.001$; $0.88 \pm 0.22$ mm y$^{-1}$) (Figure 2). Accretion varied both spatially and temporally within the *Sarcocornia* and *Sporobolus* assemblages. Across *Sporobolus* plots, there was relatively high accretion recorded at the 11-month interval, followed by multiple peaks and troughs in the amount of height of material measured above MHs, with some similarity among replicate plots in the timing of these (Fig 2b). After modest gains at the 11-month interval, *Sarcocornia* accretion diverged among plots with two plots (*Sarcocornia* 2 and 5) experiencing continued accretion, whilst *Sarcocornia* 3 and 4 appeared to lose surface material through the remainder of the study. The





pattern of accumulation and loss observed between 13-19 months at *Sarcocornia* 1 was mirrored in the nearby *Sporobolus* 1 plot.

### 3.2 Short-term deposition

#### 3.2.1 Vials

Mean bulk material deposition rates as determined by vials were higher than filter bulk deposition rates across all sampling events and vegetation assemblages (Table 1). Observations of materials retained within vials suggested a dominance of mineral matter and unidentified detritus, except in *Juncus* plots where *Juncus kraussii* fragments were the dominant material.

Deposition varied significantly among tidal events ($F_{6, 42} = 10.01$; $P < 0.001$), with post-hoc tests revealing each

event as significantly different to the others. Despite large differences in mean deposition among the three vegetation assemblages during December spring, January neap and January spring events (Table 1), vegetation assemblage was not a significant factor when elevation was included as a covariate ($F_{2, 45.8} = 1.06$; $P = 0.36$). There was, however, a significant event × assemblage interaction ($F_{6, 42} = 10.01$; $P < 0.001$), with deposition in *Sarcocornia* vials higher during January neap relative to December spring for *Sarcocornia* plots, but not so for

*Sporobolus* and *Juncus* vials (Table 1). Deposition into vials was lowest for all three assemblages during December neap (Table 1) and was highest overall in *Sarcocornia* vials during January spring (100.78 ± 32.73). Regression of the log (mass of material retained within vials) versus plot surface elevation revealed no clear relationship between the two variables during the December neap period (Fig. S1a), but significant negative relationships ($P<0.001$, $R^2>0.35$) existed for all other time periods (Fig. S1b-d). That is, there were broad trends

of higher sedimentation at lower elevation plots than higher elevation plots during these periods.

#### 3.2.2 Filters

Retention of bulk materials on filters also varied among all four tidal periods ($F_{3, 109.3} = 48.82$; $P < 0.001$), with overall deposition highest in January spring, followed by December spring (Table 1, Fig. 3). Bulk deposition on filters varied among vegetation assemblages ($F_{2, 30.85} = 48.82$; $P = 0.004$), with lower deposition in *Sporobolus*

plots relative to both *Sarcocornia* and *Juncus* plots across all tidal events (Fig. 3; Table 1). In contrast to the vials, there was no clear relationship between bulk material retained on filter papers and plot surface elevation during either of the neap or spring tidal events (Fig. S2).

Although the mass of bulk material retained on filters was similar across *Sarcocornia* and *Juncus* plots, Fig. 3 demonstrates that different materials were contributing to surface accumulation among the two vegetation

assemblages. In *Juncus* plots, autochthonous plant litter (that is, from the dominant species *Juncus* kraussii and the sub-dominant species *Sporobolus* virginicus) contributed between 66% (December neap) and 78% (both December spring and January neap) of all deposited material. In contrast, litter contributions were low (≤ 12% of all deposited material) in both *Sarcocornia* and *Sporobolus* assemblages, regardless of tidal period. Contributions from identifiable allochthonous materials were low in all cases, with negligible quantities of *Posidonia australis*

litter (recorded in five out of all 60 *Sporobolus* filters) and a single large piece of *Hormosira banksii* deposited on a *Sporobolus* filter during December spring – the latter was considered an outlier and was therefore excluded from Fig 3.





Chemical analysis of the unidentified portion of material deposited on filters also highlights differences between the vegetation assemblages. The organic content (%C, %N) of unidentified material pooled across *Juncus* plots was much higher than for the other assemblages (Table S2), with this difference also apparent in the disparity between C accumulation rates in *Juncus* versus *Sarcocornia* and *Sporobolus* assemblages (Table 1).

**3.3 Elemental and isotopic ratios**

Elemental C:N ratios and $\delta^{13}$C values of plant biomass, litter and unidentified residues are presented in Table 2. The biomass and litter samples of the C4 grass *Sporobolus* were more enriched in $^{13}$C relative to those of the C3 species *Sarcocornia* and *Juncus* (Table 2). This distinction, however, was not as great for the unidentified residue samples, with $\delta^{13}$C values from all assemblages sitting between the $\delta^{13}$C values of the C3 and C4 saltmarsh plants.

Outputs from the isotope mixing model (Table 1) highlighted differences in source contribution among the vegetation assemblages. *Sarcocornia* residues showed a higher contribution of C3 plant material during spring tides relative to the neap tides. Further, similar contribution from the host plant (i.e. C3 in *Sarcocornia* and C4 in *Sporobolus*) to residues was apparent for all tidal periods except January neap, when the C4 contribution to *Sporobolus* residue was higher. Overall, contributions of the host plant ranged from 59.6 – 77.5 % in *Sarcocornia*

plots and 61.7 - 80.1% in *Sporobolus* plots.

Source contributions across the four tidal periods were most consistent in the *Juncus* assemblage, where estimates ranged between 78.8% and 84.6% for C3 plant material and 15.4 - 21.2% for C4 plant material. These contributions aligned well with visual observations of plant cover across plots (where the C3 plant *J. kraussii* is dominant over the C4 plant *S. virginicus* in approximately an 80:20% biomass mix). Quantification of litter fall

onto filters, however, highlighted a skew towards *J. kraussii* litter (85.9 – 97.0%) over *S. virginicus* litter (3.0 – 14.1%) across the *Juncus* assemblage. Residue C:N ratios were also highest for *Juncus*, followed by *Sporobolus* then *Sarcocornia*. While *Juncus* litter samples had a higher C:N, relative to all other *Juncus* biomass (Table 2), this difference was not noted for *Sarcocornia* nor *Sporobolus*.

**3.4 MIR and $^{13}$C NMR**

Together, the first two principal components explained 96.4% of the variation in MIR spectra of all samples assessed. A clear separation of residue samples from litter and biomass is apparent along PC1 (Fig 4A) with inspection of the loadings plot (Fig 4B) highlighting variation in the range 600-2000 cm$^{-1}$ (quartz), and distinct

troughs at 3400 cm$^{-1}$ (water) and 2900 cm$^{-1}$ (OM-alkyl). Residue samples are separated along PC2, with differentiation among vegetation assemblages, regardless of tidal event. The loadings plot for PC2 (Fig 4C) also exhibits variation in the range 600-2000 cm$^{-1}$ (quartz), a peak at 2900 cm$^{-1}$ (OM-alkyl) and also 3600-3700 cm$^{-1}$ (kaolinite).

The proportions of C within each of eight organic functional groupings for each sample analysed with $^{13}$C NMR

are presented in Table 2. For all samples O-Alkyl C was the most abundant. O-Alkyl C content was higher in live plant biomass than litter for both *Sarcocornia*, but less so for *Sporobolus* and essentially unchanged for *Juncus*. Generally, residues were higher in Alkyl C, and Amide/Carboxyl C, and lower in O-Alkyl, Di-O-Alkyl and aromatics relative to litter and biomass samples. There were differences in residue C composition according to which vegetation assemblage they were collected from –aromatics (higher in *Juncus* and *Sporobolus*), Alkyl C





and Amide/Carboxyl (higher in *Sarcocornia*). There was high similarity between residues collected under the two different tides, however, for both the *Sarcocornia* and *Sporobolus* assemblages. These similarities among tides are mirrored in the similarity of the residue C:N values. There was insufficient residue material available for analysis from *Juncus* neap tide, even though samples were pooled across a large number of filters, further

highlighting the small contribution of unidentifiable sedimentary components within this assemblage.

## 4 Discussion

In this study we have compared sediment and C accretion dynamics among three vegetation assemblages within an intertidal wetland complex. Our findings, across a range of methods, showed that there were substantial differences among assemblages in: 1) the types of materials deposited on the marsh surface; and 2) the quantities

of material accumulated over 19 months. Here, we first consider the accumulation differences among assemblages over the medium-term, and then discuss the interactions among vegetation, physical and degradation processes which are likely driving these differences. We conclude with an assessment of the implications for C accumulation and storage, and response to relative sea level rise (RSLR).

### 4.1 Accretion varies among vegetation assemblages

Surface accretion above feldspar MHs over a period of 19 months and deposition within short-term sedimentation traps provide evidence of the multiple ways in which accretion dynamics differ between saltmarsh vegetation assemblages. First, feldspar MHs highlight a record of continued and consistent accretion across the upper marsh *Juncus* assemblage, amounting to a reliable ($R^2 = 0.68$) accretion rate of $1.74 \pm 0.13$ mm y$^{-1}$ (Fig. 2). This value is remarkably similar to the mean accretion rate measured over 10 years above feldspar MHs of 1.76 mm y$^{-1}$ by

Saintilan et al. (2013) for *Juncus* saltmarshes across a range of sites in SE Australia. In contrast, accretion above MHs in *Sarcocornia* and *Sporobolus* assemblages varied substantially– both spatially and temporally – in our study (Fig. 2), possibly due in part to the influence of bioturbation and sediment mixing above MHs (Cahoon and Turner, 1989; Krauss et al., 2003). While our mean accretion estimates for both *Sarcocornia* and *Sporobolus* are lower than the regional estimate for *Sarcocornia*/*Sporobolus* associations ($1.11 \pm 0.08$ mm y$^{-1}$) (Saintilan et al.,

2013), this regional mean is within the 95% confidence interval for both species at Towra Point (Table 1). Critically, accretion rates in the *Juncus* assemblage consistently exceed contemporary rates of sea level rise within Botany Bay (1.15 mm y$^{-1}$), while mean accretion rates for both *Sarcocornia* and *Sporobolus* (and even the upper 95% confidence interval of *Sarcocornia)* are below the contemporary rate of sea level rise.

### 4.2 Processes driving spatial variability in deposition and accretion

One of the key strengths of using short-term accumulation methods is the ability to identify and quantify the composition of inputs which may be contributing to differences observed over the medium-term. In this study, a distinction was observed between the organogenic deposition which dominated the *Juncus* assemblage (where medium-term accretion rates were consistently high) and the minerogenic deposition of the *Sarcocornia* and *Sporobolus* assemblages (where medium-term accretion rates were lower and more varied). This distinction was

best exemplified by the results of the filter method (Fig 3.), where differences in the contributions of autochthonous litter and the residual sediment (comprising mineral and organic residue components) were stark.



There was further evidence of this in the vial results, where mineral-biased deposition was high in the lower elevation, non-rush assemblages, and low in the higher elevation *Juncus* assemblage during multiple experimental periods (Table 1; Fig. S1). Although higher than predicted tides likely influenced some short-term traps during neap experimental periods (Table S1), the fact that deposition into vials was lower during December neap (when

up to 80% of *Sarcocornia* plots and 100% of *Sporobolus* plots would have been subjected to at least one tidal inundation) relative to January neap (up to 60% of *Sarcocornia* plots; no inundation of *Sporobolus* plots), suggests that this had a small impact relative to other influences.

Together, the vegetation assemblage scale differences in short-term deposition and longer-term accumulation patterns observed in this study suggest further consideration of the biological, physical and interactive processes

which are most responsible for the dynamics of saltmarsh surface materials is warranted.

### 4.2.1 The role of vegetation

There are fundamental differences in vegetation structure and function which can at least partly account for the variations in the quantity and type of materials being retained in rush (*Juncus*) versus non-rush (*Sarcocornia* and *Sporobolus*) assemblages. First, *Juncus* assemblages have massive potential for direct organic sedimentation

through the annual replacement of their significant aboveground biomass (1116 g m$^{-2}$) (Clarke and Jacoby, 1994). No clear patterns of annual turnover have been observed in *Sarcocornia* and *Sporobolus* assemblages, where standing biomass is only about one-third that of the *Juncus* assemblage (Clarke and Jacoby, 1994).

There may also be indirect vegetation effects on the deposition and accumulation of surface materials. For instance, the tall (~1 m), dense structure of the *Juncus* assemblage is likely to enhance: 1) the retention of

autochthonous litter which may have otherwise been exported during tidal recession, and 2) the capture of mineral particles on plant stems (Morris et al., 2002; Mudd et al., 2004). Dense saltmarsh vegetation also has the capacity to enhance sedimentation by reducing the turbulent energy of inundating waters, with Mudd et al. (2010) demonstrating that this phenomenon was responsible for virtually all of the sedimentation increase observed when standing plant biomass of *Spartina alterniflora* was artificially increased. The high litter deposition rates we

observed during neap tides (Fig. 3a,c) and the increased contribution of both mineral and litter components during spring tides (Fig 3b,d) suggested that each of these direct and indirect plant-mechanisms may be operating and contributing to the relatively high medium-term accretion rates within *Juncus* assemblages. This supported the first element of our first hypothesis - that assemblage differences can be (at least partly) explained by differences in vegetation structure.

### 4.2.2 The role of physical factors

Differences in suspended sediment supply and tidal flooding characteristics (tidal range, position within the tidal prism) have been identified as key physical drivers of saltmarsh accretion (Chmura and Hung, 2004; Rogers et al., 2014). Generally, lower elevation within the tidal frame and closer proximity to the source of tidal inundation result in higher sedimentation rates. This is because: (1) greater flooding depth allows for greater sediment volume

and higher sedimentation, and (2) the increase in flooding duration increases the time for sediment deposition to occur (Baustian et al., 2012; Harter and Mitsch, 2003; Morris, 2007; Oenema and DeLaune, 1988). If these processes were operating in our site, we would have expected to observe higher sedimentation rates in the *Sarcocornia* and *Sporobolus* assemblages, which were generally both lower in the tidal frame (Table S1) and





nearer to tidal sources (Fig. 1). Indeed, when measurements relevant to the mineral component were considered, our results appeared to be consistent with this. First, overall mineral retention on filters (Fig. 3) was highest in the *Sarcocornia* and *Sporobolus* assemblages. Second, mineral-biased deposition into vials was shown to have a significant log-linear relationship with elevation during the periods of greatest tidal inundation (December spring,

January spring), and during January neap when significant rainfall (Fig. S3) as well as some inundation of low elevation sites likely occurred (Table S2). These mineral deposition results were therefore supportive of the role of physical position within the saltmarsh towards differences among assemblages (i.e. part two of our first hypothesis).

Importantly, however, the deposition-elevation relationship expressed by the mineral component, did not apply

when bulk results of the passive filter method were considered. With the mineral bias effectively removed, no clear relationship between elevation and bulk deposition was observed across any of the tidal periods (see Fig. S2). Instead, total deposition was similar between the minerogenic, lower elevation *Sarcocornia/Sporobolus* plots, and the organogenic, higher elevation *Juncus* plots.

The lack of an elevation relationship in terms of bulk material deposition is somewhat contrary to spatial patterns

expected on the basis of physical sedimentary processes in the tidal zone. This disparity extended to medium-term accretion results, where lower marsh (*Sarcocornia* and *Sporobolus*) assemblages accrete at a slower rate than upper marsh (*Juncus*), both in our study, and regionally (Saintilan et al., 2013). This relationship doesn't necessarily downplay the importance of tidal influence on surface dynamics in SE Australian saltmarshes. An alternative explanation is that these physical processes, in interaction with biological factors, are instead

remobilising and redistributing materials across the lower marsh assemblages, rather than depositing significant amounts of 'new' allochthonous material.

### 4.2.3 Redistribution of surface materials

The second hypothesis of our study was that the source and character of materials deposited would vary temporally with tidal inundation patterns. For the most part, however, this was not observed, with high degrees of within-

assemblage similarity for neap and spring tide samples across the various analyses undertaken (Table 2; Fig. 4). Instead, our results provide multiple lines of evidence that point to the redistribution of surface materials across the saltmarsh, mediated by a range of biological, physical and interactive processes.

The first indication of redistribution of surface materials was the mismatch between rates of short-term bulk deposition and patterns of medium-term accretion among vegetation assemblages. This was best exhibited in the

*Sarcocornia* plots, where short-term measures showed deposition to be as high or higher in *Sarcocornia* plots relative to the other assemblages (Table 1; Fig. 3), while medium-term accretion was actually lowest here (Fig. 2). This suggests that short-term measures in this assemblage were capturing materials which were being moved or redistributed across the saltmarsh, but not necessarily retained in a given location over longer time periods (i.e. months). While the short- versus medium-term discrepancy was not as large for the *Sporobolus* assemblage, the

temporal variability in feldspar MH measurements (i.e. multiple peaks and troughs across the 19 month period for most plots) also suggested significant redistribution of materials over time in this assemblage. Such movement of materials within the *Sarcocornia* and *Sporobolus* assemblages also fits with the expectation that hydrodynamic energy, and therefore potential for sediment redistribution, would be highest in the saltmarsh zones lower in the tidal frame and located closer to tidal sources (Fig. 1, Table S1). We also attribute the fading of feldspar horizons





in many *Sarcocornia* and *Sporobolus* plots over time to mixing of sediments (Cahoon and Turner, 1989) in this active zone, with assistance from bioturbation (Cahoon and Turner, 1989; Krauss et al., 2003). In contrast, these temporal discrepancies and variations (including fading of MHs) were not observed in the *Juncus* assemblage, where hydrodynamic energy is expected to be greatly reduced as a result of both its position within the marsh and

the influences of plant biomass (see discussion above).

Next, it was not expected that tidal inundation would substantially increase saltmarsh plant litter production. We therefore interpret the increased concentration of autochthonous litter in *Juncus* plots during spring tides relative to neap tides (Fig. 3) as evidence of the redistribution and trapping of autochthonous material within this assemblage. That is, the 'extra' spring tide litter was material that had been remobilised by inundating water and

redistributed within the same community, resulting in a larger amount of material being caught on the *Juncus* filters. The fact that no identifiable *Juncus* litter was collected on any of the *Sporobolus* filters, despite their position being within the expected path of receding tides (Fig. 1), further highlights the retaining capacity within the *Juncus* assemblage. While it is not known over what scale the litter redistribution is occurring in the *Juncus* assemblage, we expect it to be highly localised, given the dense structure of standing vegetation here and its

capacity to impede movement of coarse litter particles.

Finally, by placing our *Sarcocornia* and *Sporobolus* plots within small patches vegetated exclusively by either the C3 species (*S. quinqueflora*) or the C4 species (*S. virginicus*), we are able to estimate the contribution of each resident plant to the residue collected from within its assemblage. While the dominance of resident plant signatures suggested a strong autochthonous contribution in all instances (see mixing model results in Table 1), residue

signatures across all tidal periods reveal a mixture of sources both present (i.e. the resident plant) or neighbouring (i.e. the other co-dominant plant in the association) to the plots. The fact that contributions of sources other than the resident plant were in the order of 20 – 40% (Table 1) during the neap tides suggest significant mixing across scales greater than the monospecific patches (i.e. several metres or more). While some of this movement of materials may have been due to the creep of the highest neap period tides into the lower elevation plots (though

this appears small - see section 4.2), non-tidal agents such as redistribution by rainfall (Chen et al., 2015) and faunal activity (Guest et al., 2004) may have also contributed.

A two source (C3 plant v C4 plant) mixing model probably presents an overly simplified estimate of source matter contributions. This is because it does not account for other potential sources which have $\delta^{13}C$ values within or near the range of saltmarsh plant sources prescribed in the mixing model. These include mangroves (-28.7 ± 0.3

‰), seagrass (-12.3 ‰), macroalgae (-17.7 ‰) and benthic algae (-15.0 ± 0.4 ‰). Of these, benthic algae would have the greatest potential to contribute to *Sarcocornia* residue, as vegetation is sparsest here (and therefore light penetration to benthos the greatest), while the MIR PC plot (Fig. 4) also points to a similarity in chemical composition between the two. However, the fact that *Sporobolus* residues are consistently depleted in $^{13}C$, relative to both the resident plant (*S. virginicus*) and benthic algae, show that our interpretation of mixing between both

C3 and C4 sources is warranted at least in that assemblage. In contrast, the constancy of isotope signatures and their overall similarity with the mix of C3- and C4-derived biomass in the *Juncus* plots provide further evidence of the autochthonous nature and trapping capacity of this assemblage.

Together, these findings allow several conclusions to be made about redistribution of surface materials. First, short-term deposition measures may capture a significant proportion of within-marsh redistribution and therefore

may not necessarily equate with longer term accretion. Second, the capacity of vegetation to retain autochthonous





sedimentation appears to vary substantially among species assemblages. Third, redistribution is likely to be greatest in more exposed, lower-biomass assemblages. These findings also highlight the importance of considering redistributed materials in quantifications of wetland surface dynamics, and likely shortcomings for studies which attempt to assess surface dynamics using only short-term methods.

**4.3 Implications for wetland function**

Understanding the biological and physical feedbacks which affect surface dynamics is critical to the survival of intertidal wetlands and their associated ecosystem services, under changing environmental conditions (Kirwan and Megonigal, 2013). To this end, the data collected as part of this study reveal patterns of how C sequestration capacity, organic matter decomposition and vulnerability to sea level rise vary among saltmarsh assemblages.

**4.3.1 C deposition and sequestration rates**

The distinction between organogenic and minerogenic assemblages, and their respective locations within the tidal frame, has important implications for surface C sequestration rates. Here we estimate mean C deposition rates ranging from 0.03 to 0.23 g C cm$^{-2}$ y$^{-1}$ across the four tidal periods for the minerogenic *Sarcocornia* and *Sporobolus* assemblages and 0.41 to 0.87 g C cm$^{-2}$ y$^{-1}$ for the organogenic *Juncus* assemblage (Table 1). It should be noted that such short-term C deposition rates inclusive of plant litter will likely represent a massive overestimation of C that is retained and sequestered over longer timescales, due to diagenesis of deposited OM (Duarte and Cebrian, 1996), and the potential for materials to be redistributed or even exported by tidal and non-tidal processes (see section 4.2.3). Therefore, these deposition rates are not directly comparable to C accumulation rates determined by medium-term (e.g. feldspar MH) or longer term (e.g. radiometric dating) techniques. Notwithstanding this, the magnitude of the differences we report among assemblages above fit broadly with differences in regional estimates of C accumulation over the medium-term (10 y MH experiments) which have been estimated as 4.5 times higher in *Juncus* relative to *Sarcocornia-Sporobolus* saltmarsh (Saintilan et al., 2013). Similarly, our results are also in agreement with findings further north in Moreton Bay, where Lovelock et al. (2013) reported much higher C sequestration rates on oligotrophic sand island marshes dominated by *J. kraussii*, than *S. quinqueflora* dominated marshes on the western side of that bay.

**4.3.2 Decomposition of organic matter varies among assemblages**

We have assessed the chemistry of aboveground biomass, litter and unidentified residues through elemental (C:N) and spectrometric (MIR, $^{13}$C NMR) methods. Together, these analyses have revealed insights into the fate of aboveground organic matter and the likelihood of their contribution to longer-term sedimentary carbon stocks. Most importantly, our results highlight among assemblage differences in the transformation of OM along the biomass-litter-sediment decay continuum.

Shifts in the bulk composition of materials was best seen in the principal components plots of MIR spectra, where biomass, litter and sediment residue samples varied across PC1 (Fig. 4a). Broadly, the separation of residues from litter and biomass was primarily due to the addition of mineral components in the residues, however, there was also evidence of a shift in alkyl OM. Specifically, the presence of a single peak at ~2900 cm$^{-1}$ in the loadings plot (Fig. 4b) was indicative of a declining cellulose content across PC1, that is, in the general order live biomass – litter – residue. Importantly, cellulose also appears to be a factor in the separation of residues from the three





different saltmarsh assemblages along PC2 (Fig. 4c), suggesting higher content in the two *Juncus* samples, followed by *Sporobolus* and then *Sarcocornia* samples. This finding was confirmed by [13]C NMR data, which showed greater proportions of plant compounds (carbohydrates more broadly, as well as lignin) were retained within the *Juncus* litter and residue relative to the other species (Table 2). In contrast, the higher proportions of

alkyl–C and amide/carboxyl–C within *Sarcocornia* and *Sporobolus* residues were indicative of higher protein and lipid contents, consistent with bacterial biomass and marine algae signatures (Dickens et al., 2006). However, they may also be partly explained by the selective retention of resistant plant waxes, such as suberin and cutan. There are multiple mechanisms which may explain the greater retention of plant-derived C along the biomass-litter-residue pathway for *Juncus*, relative to the other assemblages. The simplest explanation is that a high

turnover of *Juncus* biomass (and its exclusion of other sources through shading and/or structural impedance) ensures ample supply of plant-derived C to the benthos. Our data, however, reveal an important biomass-to-litter transformation in *Juncus* that was not observed in either the *Sarcocornia* or *Sporobolus* assemblage. That is, the C:N of *Juncus* litter increased substantially relative to live biomass. Such an increase is commonly observed in terrestrial (McGroddy et al., 2004) and marine (Stapel and Hemminga, 1997) plants and may be explained by the

selective resorption of nutrients (but not carbon) by the plant prior to, or during, senescence (McGroddy et al., 2004; Stapel and Hemminga, 1997). Such a mechanism was supported by the constancy of molecular C composition between *Juncus* biomass and litter (Table 2). The selective resorption of N by a plant has important implications for the fate and processing of the resulting litter and residue, as tissue C:N is considered a primary determinant on saltmarsh organic matter decomposition (Minden and Kleyer, 2015). By retaining nutrients within

the living tissues, the plant effectively decreases the lability of resulting litter and residual sediments and makes them less attractive to the microbial decomposer community (Reddy and DeLaune, 2008). This will have the effect of lowering OM remineralisation rates in *Juncus* relative to other assemblages, a result which also coincides with the bacterial biomass increases suggested for *Sarcocornia* and *Sporobolus*, but not apparent within the more recalcitrant *Juncus* residues (Table 2). Finally, there may also be an element of physical protection, with the closed

structure of the *Juncus* assemblage potentially offering increased protection against decomposition with lower, more stable temperatures expected at ground level, relative to the more exposed *Sarcocornia* and *Sporobolus* assemblages.

Together, these data from SE Australia contribute to a broader pattern of plant assemblage differences in saltmarsh surface dynamics and C sequestration potential (Minden and Kleyer, 2015; Saintilan et al., 2013; Wang et al.,

2003). They also highlight the likely processes behind the high capacity of *Juncus* saltmarshes to accumulate significant C stocks globally (0.034 g C cm$^{-2}$ y$^{-1}$), relative to most other saltmarsh genera (mean C accumulation rate = 0.024 g C cm$^{-2}$ y$^{-1}$) (Ouyang and Lee, 2013).

### 4.3.3 Vulnerability to sea level rise

There is growing evidence of the capacity of coastal wetlands to maintain surface elevation with relative sea level

rise (RSLR), in certain situations, by increasing surface elevation through belowground production, enhanced trapping of sediments, or a combination of the two (Baustian et al., 2012; Kelleway et al., 2016b; McKee et al., 2007). Where wetland assemblages are unable to maintain a suitable elevation relative to inundating water levels then vegetation shifts may occur, including the loss of marsh vegetation (Day Jr et al., 1999; Day Jr et al., 2011; Rogers et al., 2006). While wetland surface elevation is a function of multiple factors, including belowground





production and decomposition, groundwater dynamics and sedimentary and regional subsidence (Cahoon et al., 1999; Rogers and Saintilan, 2008), the retention of aboveground inputs play a critical role in wetland survival under changing hydrological conditions (Day et al., 2011).

With this in mind, our medium-term accretion data suggest that *Sporobolus* and *Sarcocornia* assemblages may be
particularly vulnerable to current RSLR, with mean surface accretion rates either lower (*Sarcocornia* = 0.92 mm y$^{-1}$) or only marginally higher (*Sporobolus* = 1.30 mm y$^{-1}$) than contemporary rates of local sea level rise within Botany Bay (1.15 mm y$^{-1}$). In fact, there is already evidence of this across much of the Towra Point Nature Reserve, as well as elsewhere in the region, where upslope encroachment of mangrove shrubs into *Sarcocornia-Sporobolus* association is occurring (Kelleway et al., 2016b). In contrast, vegetation change (either in the form of
mangrove encroachment or dieback) has not been widely reported for *Juncus* assemblages across SE Australia over recent decades, suggesting relative stability during a time of changing sea levels. While belowground biomass production likely plays a role, average *Juncus* surface accretion rates (1.70 mm y$^{-1}$ in this study; 1.76 mm y$^{-1}$ regionally) in excess of local RSLR suggest a potential role of aboveground inputs towards maintaining surface elevation. Dependence upon organogenic inputs for accretion, however, also means the response of *Juncus*
assemblages to RSLR may vary with shifts in productivity or decomposition dynamics (e.g. changes in climate and/or nutrient status). Under present conditions, at least, our analyses have shown these organic inputs to be relatively resistant to early decomposition. In all, our findings are also supportive of recent research which suggests organic sediment accretion may be of critical importance in marsh survival under RSLR, particularly in areas most removed from inorganic sediment delivery (D'Alpaos and Marani, 2015). Whether belowground
organic matter production makes substantial contributions to Australian saltmarsh surface elevation dynamics and vulnerability to sea level rise remains unknown, and represent an important area for further research. Better understanding of the temporal dynamics of organic and mineral contributions to elevation maintenance is also required, including in relation to expected non-linear increases in sea level.

By combining medium-term accretion quantification with short-term deposition measurements and chemical
analyses we have gained insights into the various processes behind observed differences in accretion among saltmarsh vegetation assemblages. While our study highlights assemblage scale differences in potential response to RSLR, it represents only a small part of the information needed to accurately predict the future of SE Australian saltmarsh assemblages. Further measures of short-term deposition and medium-term accretion across a broader range of geographical settings, longer-term studies of soil elevation change among assemblages and modelling of
vegetation response thresholds are all required.

**Acknowledgements**

Charlie Hinchcliffe, Mikael Kim, Sarah Meoli and Frederic Cadera assisted with field measurements. Janine McGowan and Bruce Hawke assisted with sample preparation and spectroscopic methods. Field collections were undertaken in accordance with NSW Office of Environment and Heritage scientific licence SL101217 and NSW
Department of Primary Industries Scientific Permit P13/0058-1.0. We also thank NSW National Parks and Wildlife Service for supporting access to Towra Point Nature Reserve. This research was supported by the CSIRO Coastal Carbon Cluster. PM was supported by an Australian Research Council DECRA Fellowship (DE130101084). Data from this research will be made public as part of the CSIRO Coastal Carbon Cluster.



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

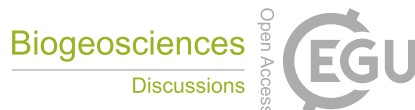

**Table 1.** Summary of sediment measure techniques; mean and standard error values for feldspar marker horizon (MH), vial and filter bulk sediment measures; mean and standard error of C deposition rates; and contributions of C3 and C4 vegetation to deposition among vegetation assemblages. Accretion rates slower than local sea level rise (1.15 mm y⁻¹) are in red. DN = December neap; DS = December spring; JN = January neap; JS = January spring.

| Technique | Parameter *Measure* Biases/issues | Period | *SARCOCORNIA* | | | *SPOROBOLUS* | | | *JUNCUS* | | |
|---|---|---|---|---|---|---|---|---|---|---|---|

**Feldspar MH**

| | **Bulk measure** *sediment accumulation* mid-term (days to years) MH may be lost through erosion or bioturbation | | accretion rate (mm y⁻¹) | | | | | | | | |
|---|---|---|---|---|---|---|---|---|---|---|---|
| | | | mean | | SE | mean | | SE | mean | | SE |
| | 0–19 mo. | | 0.78 | | 0.18 | 0.88 | | 0.22 | 1.74 | | 0.13 |
| | 95% CI lower | | 0.42 | | | 0.44 | | | 1.48 | | |
| | 95% CI upper | | 1.14 | | | 1.32 | | | 2.00 | | |
| | | | *see also Figure 2* | | | | | | | | |

**Vial**

| | **Bulk measure** *sediment deposition* short-term (hours to days) Biased towards materials entrained by water. Biased against coarse litter larger than vial mouth | | bulk deposition rate (g cm⁻² y⁻¹) | | | | | | | | |
|---|---|---|---|---|---|---|---|---|---|---|---|
| | | Period | mean | | SE | mean | | SE | mean | | SE |
| | | DN | 3.41 | | 1.14 | 2.74 | | 0.90 | 3.79 | | 1.25 |
| | | DS | 25.19 | | 3.88 | 66.52 | | 10.47 | 18.13 | | 3.26 |
| | | JN | 38.67 | | 9.20 | 18.84 | | 5.49 | 4.99 | | 1.33 |
| | | JS | 100.78 | | 32.73 | 80.94 | | 8.92 | 14.92 | | 2.14 |

**Filter**

| | **Bulk measure** *sediment deposition* short-term (hours to days) | | bulk deposition rate (g cm⁻² y⁻¹) *see Figure 3* | | | | | | | | |
|---|---|---|---|---|---|---|---|---|---|---|---|
| | | Period | mean | | SE | mean | | SE | mean | | SE |
| | | DN | 1.47 | | 0.19 | 1.09 | | 0.26 | 1.24 | | 9.7E-04 |
| | | DS | 2.18 | | 0.35 | 1.31 | | 0.10 | 2.37 | | 2.5E-03 |
| | | JN | 0.79 | | 0.20 | 0.43 | | 0.05 | 1.02 | | 2.0E-03 |
| | | JS | 3.04 | | 0.79 | 2.50 | | 0.46 | 2.55 | | 2.8E-03 |

| | **Filter + material identification** *composition of material deposited* | | | | | | | | | | |
|---|---|---|---|---|---|---|---|---|---|---|---|

| | **Filter + elemental analysis** *C deposition rate* | | C deposition rate (g C cm⁻² y⁻¹)ᵃ | | | | | | | | |
|---|---|---|---|---|---|---|---|---|---|---|---|
| | | Period | mean | | SE | mean | | SE | mean | | SE |
| | | DN | 0.10 | | 0.02 | 0.08 | | 0.02 | 0.44 | | 0.06 |
| | | DS | 0.09 | | 0.02 | 0.12 | | 0.02 | 0.87 | | 0.16 |
| | | JN | 0.03 | | 0.01 | 0.06 | | 0.01 | 0.41 | | 0.15 |
| | | JS | 0.23 | | 0.03 | 0.15 | | 0.03 | 0.73 | | 0.17 |

| | **Filter + isotopic analysis** *sources contributing to sediment residue* | | isotope mixing model – plant contribution (%) | | | | | | | | |
|---|---|---|---|---|---|---|---|---|---|---|---|
| | | Period | C3 | | C4 | C3 | | C4 | C3 | | C4 |
| | | DN | 59.61 | | 40.39 | 38.3 | | 61.7 | 80.2 | | 19.79 |
| | | DS | 77.50 | | 22.50 | 25.0 | | 75.0 | 84.6 | | 15.38 |
| | | JN | 67.94 | | 32.06 | 19.9 | | 80.1 | 81.0 | | 18.98 |
| | | JS | 72.39 | | 27.61 | 26.0 | | 74.0 | 78.8 | | 21.21 |
| | | | *see also Table 2* | | | | | | | | |

| | **Filter + MIR & ¹³C NMR** *character of deposited materials* | | *see Figure 4 & Table 2* | | | | | | | | |
|---|---|---|---|---|---|---|---|---|---|---|---|

ᵃ Calculated by multiplying bulk accumulation for individual plots by %C values obtained for pooled litter and residue samples





**Table 2.** Results of ¹³C NMR, δ¹³C and elemental (C:N) analyses for each component of three saltmarsh plant assemblage, plus other potential sources. ¹³C NMR outputs are the average of two samples for each of biomass and litter components (except *Sporobolus* litter, for which only one reliable spectrum was obtained), δ¹³C and C:N values are the mean (±SE) of biomass (n=3) and litter (n=4) samples. Residue samples were pooled from 15 filters in each vegetation assemblage.

| Community | Component | Tide | ¹³C NMR chemical assignment and region (ppm) | | | | | | | | δ¹³C | C:N |
| --- | --- | --- | --- | --- | --- | --- | --- | --- | --- | --- | --- | --- |
| | | | Alkyl (0 – 45) | N-Alkyl/Methoxyl (45 – 60) | O-Alkyl (60 – 95) | Di-O-Alkyl (95 – 110) | Aryl (110 – 145) | O-Aryl (145 – 165) | Amide/Carboxyl (165 – 190) | Ketone (190 – 215) | | |
| *Sarcocornia* | biomass | n/a | 14.4 | 6.6 | 45.0 | 10.6 | 10.5 | 4.4 | 7.1 | 1.4 | -26.5 ± 0.5 | 38.0 ± 4.2 |
| | litter | combined | 10.2 | 5.1 | 34.0 | 9.0 | 16.3 | 8.6 | 13.2 | 3.6 | -25.7 ± 0.1 | 33.0 ± 1.2 |
| | residue | neap | 23.1 | 8.2 | 33.7 | 7.3 | 11.3 | 4.3 | 11.0 | 1.1 | -22.1 ± 0.3 | 14.2 ± 0.4 |
| | residue | spring | 25.1 | 8.1 | 32.8 | 7.1 | 10.5 | 4.0 | 11.3 | 1.1 | -23.3 ± 0.2 | 14.2 ± 0.7 |
| *Sporobolus* | biomass | n/a | 8.6 | 4.6 | 51.0 | 12.0 | 11.5 | 4.6 | 6.5 | 1.3 | -14.9 ± 0.1 | 59.8 ± 6.3 |
| | litter | spring | 5.2 | 4.5 | 47.1 | 12.0 | 14.6 | 6.6 | 8.0 | 2.0 | -15.8 ± 0.2 | 59.4 ± 5.2 |
| | residue | neap | 19.2 | 7.9 | 36.5 | 8.3 | 12.6 | 5.0 | 9.4 | 1.2 | -18.7 ± 0.6 | 15.5 ± 0.5 |
| | residue | spring | 18.5 | 7.3 | 36.6 | 8.4 | 13.0 | 5.2 | 9.5 | 1.4 | -18.4 ± 0.0 | 16.3 ± 1.1 |
| *Juncus* | biomass | n/a | 8.6 | 5.4 | 51.8 | 12.2 | 11.2 | 4.6 | 5.2 | 1.1 | -24.8 ± 0.3 | 61.3 ± 5.6 |
| | litter | combined | 5.6 | 5.4 | 52.2 | 12.8 | 11.9 | 5.4 | 5.4 | 1.3 | -25.6 ± 0.1 | 89.9 ± 4.5 |
| | residue | neap | *insufficient material available for ¹³C NMR analysis* | | | | | | | | -23.7 ± 0.0 | 19.6 ± 0.0 |
| **Other sources** | algal mat | spring | 15.4 | 7.2 | 35.7 | 8.8 | 15.4 | 6.6 | 9.2 | 1.7 | -23.8 ± 0.2 | 18.2 ± 0.4 |
| | mangrove leaf | n/a | 14.3 | 5.6 | 37.5 | 8.7 | 11.5 | 4.9 | 14.8 | 2.6 | -15.0 ± 0.4 | 13.2 ± 0.1 |
| | seagrass wrack | spring | 16.4 | 6.4 | 40.6 | 9.1 | 12.8 | 4.8 | 8.7 | 1.1 | -28.7 ± 0.3 | 24.8 ± 1.7 |
| | macroalga wrack | spring | 11.6 | 3.9 | 33.3 | 8.0 | 14.4 | 7.7 | 16.4 | 4.6 | -12.3 | 29.2 |
| | | spring | 6.0 | 2.6 | 52.5 | 14.0 | 5.5 | 6.6 | 11.5 | 1.3 | -17.7 | 57.8 |





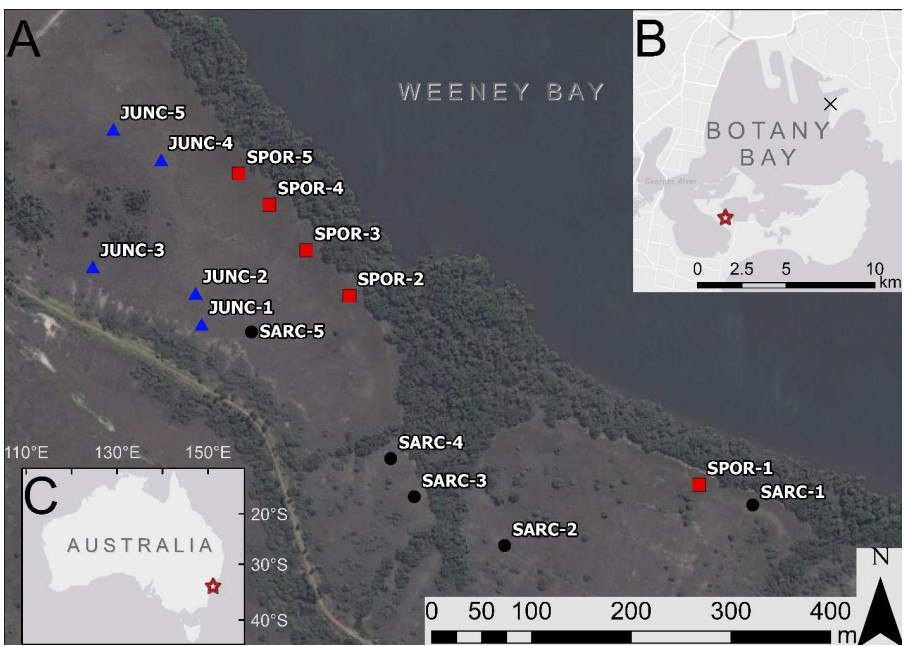

Fig 1. Location of experimental plots within Weeney Bay saltmarsh of Towra Point Nature
Reserve (A), located along the southern shoreline of Botany Bay (B) in southeast Australia
(C). Location of nearest tidal gauge is marked by an X in inset B. SARC = Sarcocornia
quinqueflora assemblage; SPOR = Sporobolus virginicus assemblage; JUNC = Juncus
kraussii assemblage.





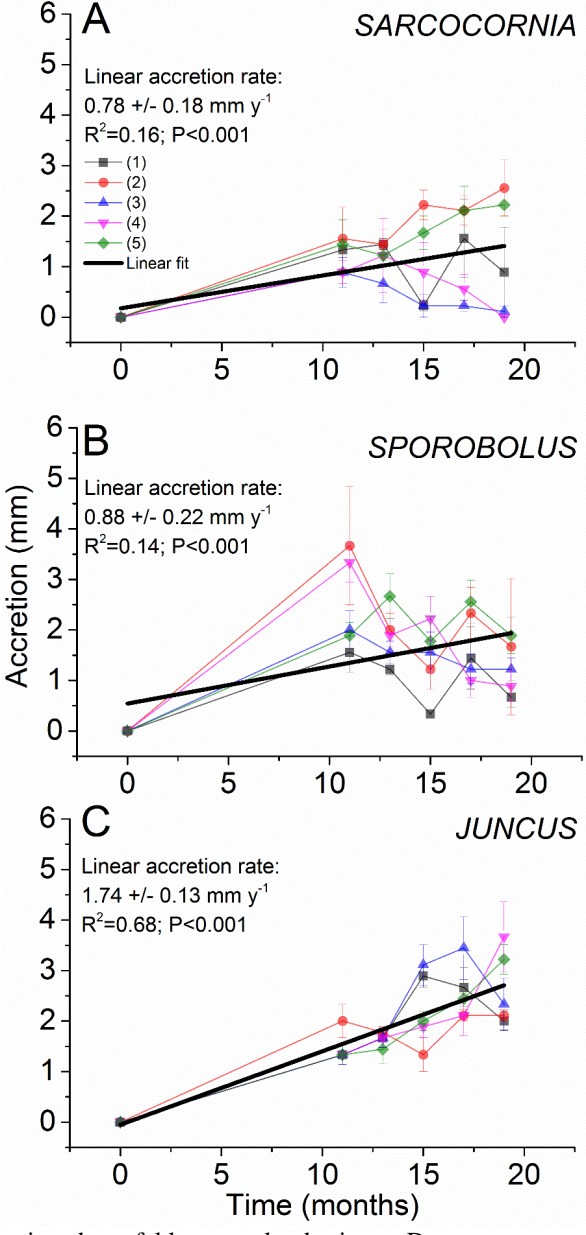

Fig 2. Surface accretion above feldspar marker horizons. Data are presented as the mean ± SE of three replicate plots in each of five locations for each vegetation assemblage. A linear fit was applied on the basis of all data points (n=90) for each vegetation assemblage.




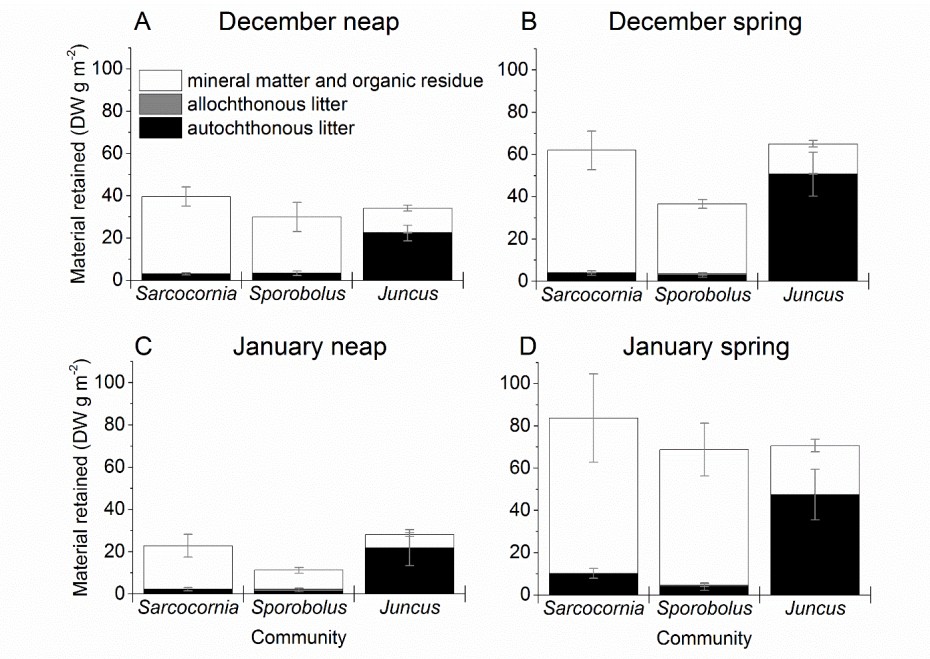

Fig 3. Mean mass of autochthonous litter, allochthonous litter and mineral matter / organic residue retained on filters at end of 6d deployment during December neap (A), December spring (B), January neap (C) and January spring (D) tidal periods. Error bars are presented for each component and represent 1 standard error each side of the mean.





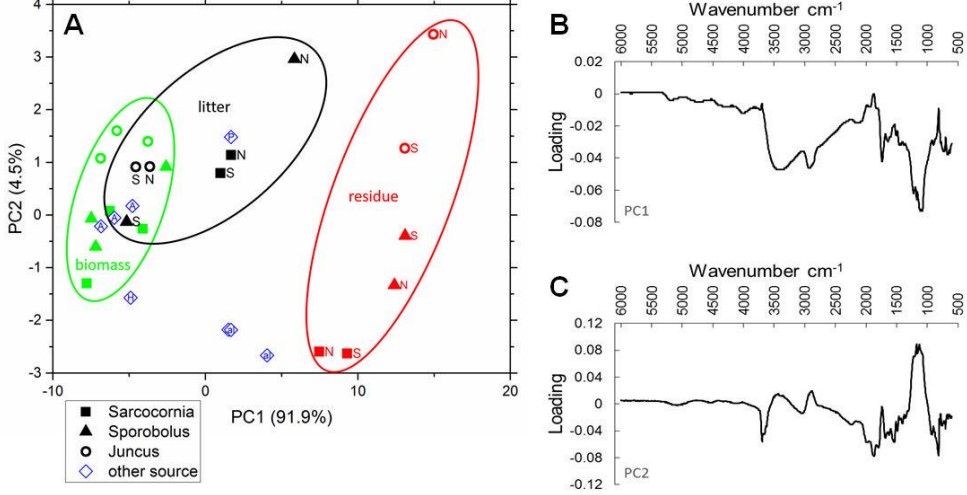

Fig 4. Principal component analysis of MIR spectra with the proportion of variance explained by each component is given in parentheses (A); spectral loading plots for PC1 (B) and PC2 (C). S = spring tide samples; N = neap tide samples; H = *Hormosira banksii* (macroalga) wrack; P = *Posidonia australis* (seagrass) wrack; A = *Avicennia marina* (mangrove) leaf; al = benthic algae mat.