# Peer review of "Sediment and carbon deposition vary among vegetation assemblages in a coastal saltmarsh"

_Biogeosciences, 2017_

## Referee Comment (RC1) · Anonymous Referee #1 · 9 Mar 2017

doi:10.5194/bg-2017-15 // Review for Kelleway et al. submitted to biogeosciences

Kelleway et al present an interesting work on medium-term and short-term accretion and deposition dynamics in different vegetation communities of a salt-marsh site on the Australian East coast. By combining different methods for measuring short- and medium-term deposition and accretion, they were able to reveal that considerable differences exist between communities with regard to accretion and organic-matter source. The manuscript presents some novel aspects on sediment and organic matter dynamics within salt-marsh systems. Unfortunately, however, I cannot recommend the work for publication before several shortcomings, often with regard to the structure of the ms, have been considered. Overall, the connection between hypotheses/research

questions and the rest of the ms is very weak. Thus, large parts of the discussion have not been sufficiently set up in the introduction and particularly not in the hypotheses. No doubt the study used interesting methodology and a wide array of tools; however, in most parts it does not become clear to the reader why certain analyses/methods were conducted or why they are necessary at all until one gets to the respective parts in the discussion of the ms. The authors need to make clear that this work is not simply about comparing different methods for assessing deposition, accumulation, and accretion dynamics. I will try to elaborate on this in the following:

NOTE: Your line numbering starts over on every new page, that was tricky ;)

Title page:

L1 The title could be more specific, but I don't have strong opinions on that. It seems that throughout the ms you rather use the terms deposition and accretion. So why is "accumulation" used in the title?

L18 Please make clear that this is a case study, conducted in one marsh system only. "within 3 vegetation types common throughout Australia" could be misleading and can give the impression that this is a larger scale study which has been replicated in several systems. Please, also discuss implications of this missing replication.

Main part: L1+3 please be consistent in your wording coastal wetland <-> coastal saltmarsh, please use different terms only if you mean different things, otherwise that can be confusing..

L13 give correct reference Kirwan instead if "Kirwin"

L18 you use the term sediment for both, the suspended matter that can deposit on the marsh surface but also to that what others refer to the "soil" of the marsh. I know, that is hair-splitting, but please make sure that you don't confuse the reader too much. Especially when you are talking about organogenic systems (L15), you should not use the term sediment when actually referring to something like a peat soil. Please

check out "Do marine rooted plants grow in sediment or soil? A critical appraisal on definitions, methodology and communication" (Kristensen and Rabenhorst 2015) for clarity.

L24-25 There is also work focused on other species (Schoenoplectus) by for instance Langley or Langley and Megonigal (PNAS or Nature) or by Rooth (2003) on Phragmites that could be mentioned here.

L39 I think this study of Kirwan et al (2013) was only on decay but not on the balance between OM inputs and decay. I think Mueller et al (2016; GCB) is more focused on the link between the two or Kirwan and Megonigal (2013; Nature) at least discusses both.

Page 3 L2 Hemminga and Buth 1991 give a nice citation here on litter-quality effects on decay

Page3 L13-15 Please give expected directions of effects in your hypotheses instead of only expecting that they will "vary".

L15-18 It seems like the second aim of this study is a methods comparison. I see this as a major weakness of the manuscript. Like mentioned above, either justify why the application of the different methods was necessary to answer you research questions or save that for a very nice second manuscript. Otherwise it is hard to follow your structure.

Page4 L4,5 give range or st deviation for biomass values

L11 to what depths was biomass assessed here?

L23+L32 Briefly mention why those measurements were conducted and don't just list them. Well, an informed reader can probably guess why you measured elevation or deployed marker horizons; however, when it comes to 2.5(isotopes) or 2.7 (13CNMR) you need to give a rationale.

Page6 L21 which functional groups, why was this done?

Page7 L17 Is this method needed to better interpret isotope data? L24 did you really assess net accretion or accumulation?

Page10 L32 why are you using "organogenic" instead of "organic" deposition?

Page12 L28-34 I think this is a real highlight of your study. Try to better set up this whole redistribution thing in your intro. I guess there is relatively little known about these dynamics.

Page13 L16-18 I don't buy that based on 13C natural abundance only! Did you consider that 13C-fractionation processes during of organic matter decay are inducing shifts in your signatures? Are differences between litter and fresh biomass large in your species? Can your other methods support/help here?

Page14 L11 following: I think it goes too far to discuss sequestration rates based on the presented data. You studied processes on the marsh surface, which may affect C sequestration, but here you should really stick to "deposition". Also "surface C sequestration" sounds odd to me. I don't know if C sequestration can be determined at the surface if a more or less permanent process is meant. It needs to become clear that deposition, accumulation, and sequestration are different processes. Further down in the paragraph you are using accumulation again. Please be sure to be consistent in the use of terminology.

L26 and the whole paragraph: You don't have a hypothesis on decomposition. This needs to be linked!

Page15 L21 "Reddy and DeLaune 2008" is a nice textbook indeed, but I know there is a bunch of peer-reviewed primary research or even review articles out there that should be rather cited here!

---

## Referee Comment (RC2) · Anonymous Referee #2 · 31 Mar 2017

*Review*

**Kelleway et al. 2017, Sediment and carbon accumulation vary among vegetation assemblages in a coastal saltmarsh**

Kelleway et al. present a study on the effect of different vegetation species on the trapping of mineral and organic deposits on a tidal marsh in southeast Australia. They use three different methods to assess deposition rates at the short (days) and medium term (months). Their study provides insights in the processes controlling both mineral sediment deposition and deposition of organic matter on a tidal marsh platform. Although their results make a substantial contribution to our knowledge of processes controlling tidal marsh growth and organic carbon dynamics in these environments, some issues need to be resolved before publication of the manuscript is possible, as I point out in my comments below.

General comments

One of my main concerns is that the authors use short-term (days) deposition data measured only in December and January to draw conclusion on longer term carbon and sediment dynamics, since they express the accumulation rates on a per-year basis. I think the authors should limit the conclusions they draw based on these data to short-term deposition rates, instead of C sequestration.

Another concern is that the authors calculate the annual sediment deposition rates using a linear regression line which does not pass through the origin to: this results in an overestimation which has to be corrected.

Furthermore, the authors compare the results of measurements at different timescales (days – months) to draw conclusions about the processes controlling accretion rates in different vegetation assemblages. However, they do not address the issues this poses, e.g. the short-term methods were only employed in December and January, so no information from the rest of the year is collected using these methods. This will have an effect on the results, and should be thoroughly addressed.

Also, as one of the aims of this paper is to compare different methods, I would like a discussion about the effect of the results obtained by these different methods on the conclusions they draw. The filter and vial methods result in C deposition rates that differ up to an order of magnitude: this is now not discussed in the manuscript and is a major shortcoming, and necessary if the authors want to use these results in order to draw conclusion based on these data.

The authors should re-consider the title they use for this manuscript, e.g. based on the work they present, the word 'deposition' could replace the word 'accumulation'.

Specific comments

P1 L24: you report the C deposition rates on a $yr^{-1}$ basis, while you only measured during 2 cycles of spring and neap tide, which is misleading for the reader. I address this issue further in my comments.

P1 L33: By stating in the abstract that you have gained novel insights into processes responsible for regional differences, you suggest that you explicitly addressed these issues at a broader regional scale, which is not the case. Furthermore, by saying '…processes *responsible* for regional differences…', you suggest that you have conclusive evidence that these processes are the most important one, which is also not the case (e.g. you didn't took belowground biomass production or

soil compaction into account). Therefore, I would change this sentence so that you make it clear you only performed these analyses for a single tidal marsh.

P1 L34-36: I would formulate this more careful, as now you imply that it is possible that belowground processes are of minor importance. This contradicts with the finding of e.g Saintilan et al. (2013) that root OC is an important component of the total OC pool in SE Australian saltmarshes, and P2 L34 of this ms.

P4 L22: I recommend to change this title to e.g. 'Surface elevation measurements'

P4 section 2.2: This method is of course characterised by substantial uncertainty. Are there e.g. no measurements of daily tidal height in the surroundings of the study area? That way you could calculate the difference between measured and predicted tidal height and use this to correct your measurements?

P5 L4: I propose you change this title to 'Sediment traps'

P5 L5-6: Here you state that the purpose is to quantify short-term deposition, while you report the measurement on a $yr^{-1}$ basis. This should be addressed (see below).

P5 L14: please explain what you mean by 'resolution'

P6 L2: The term 'residual sediment' is confusing, as this term is used to denote both residual sediments and organic matter, I propose to change this to e.g.; 'residual deposits'.

P6 L6: Jaschinski et al. (2008) is not included in the reference list

P6 L11: It's very confusing that you say here that you used MIR spectroscopy to assess the *composition* of the listed materials, as this is not done in this ms. In some cases, MIR spectroscopy is used to assess characteristics of the analysed material (e.g. C content), based on a calibration dataset, but this is not done here. I think this sentence is confusing to the reader, as you use the MIR spectroscopy results only to perform a PCA to discriminate between different types of deposits. Therefore, I would limit the materials section about MIR spectroscopy to this aspect.

P6 L14: if these procedures are important for the reader to replicate your measurements, please mention them.

P6 L15: the mid-infrared range of the electromagnetic spectrum is between $4000 – 400$ cm$^{-1}$, so why did you measure between $8000 – 400$ cm$^{-1}$? Please clarify.

P6 L15: Please clarify why the spectral range was adjusted to $6000 – 600$ cm$^{-1}$?

P6 L35-36: You say you test main and interactive effects of vegetation assemblages: please provide the effect *on what exactly*?

P6 Section 2.8: please mention that you report the confidence on the mean of replicate measurements as standard error (as I assume this is what you mean with SE). Also state how this was done an why you didn't use standard deviation to report on the spread among different replicate measurements.

P6 Section 2.8 + P7 L24-29 + section 4.1: You use a simple linear regression, which you fit through the data points representing sedimentation rates above the MH's, to obtain annual rates of sediment deposition: this technique leads to an overestimation of sediment deposition rates! As you show in figure 2: the regression lines do no pass through the origin of the graph, which implies that

after an infinitesimal timestep you have e.g. already 0.5mm accretion at the Sporobulus site. Likewise, when you use this regression line to calculate the amount of material that has been deposited after 12 months, you will overestimate this amount. This should be corrected: make sure you force your regression line to pass through the origin and calculate the deposition rates again.

P7 L9: Please explain what you mean by 'organic residue': are these the deposited macrolitter? Or all deposited materials combined? Or…?

P7 L18-19: please explain what you mean with 'composition'? It's confusing that you state that you will identify differences in composition, while you will only use PCA to plot the data on two PC's. Please better explain here how you used the PCA based on MIR spectra as an added value to standard lab analyses.

P7 L24: I don't agree that 'consistent' accretion was measured for the Juncus plots, as e.g. replica 2 remains relatively stable after 11 months and replica 1 and 3 show negative erosion rates towards the end of the measurement period. I would formulate this more careful.

P8 L12: $F_{2,45.8}$: how can the degrees of freedom of variance within groups be 45.8?

P8 L24-25: please perform a statistic to show whether the differences between sporobolus and the other vegetation types is significant.

P8 L 31: indicate if the 66% and 78% are mass percentages or some other measure?

P8 L32-37: Sarcocornia and Sporobolus plots were located on the low marsh, which are generally subject to higher water flow velocities compared to the high Juncus marsh. This can partly contribute to the lower amount of litter retained at the low marsh. Please discuss this briefly.

P10 L9: Indicate that you analysed the types of materials deposited for the short term

P10 L22: how about the effect of sediment removal through erosion?

Section 4.1: here you discuss that sedimentation rates are higher for the high marsh compared to the low marsh, which is the opposite of what is normally observed. Discuss this briefly, or refer to where you discuss this (section 4.2.2)

P10 L29: Section 4.2 has a confusing structure: in sections 4.2, 4.2.2 and 4.2.3 you describe the results from the short-term methods, while in section 4.2.1 you describe results from the long-term methods. Please indicate this e.g. in the titles of the different subsections, as this in very confusing for the reader.

P11 L6: you state here that during the January neap there was no inundation of the Sporobolus plots for the vials, but in table 2 you report deposition rates for JN in vials for Sporobolus plots. Please explain.

P11 L8-10: please better explain which 'scale differences' you mean and shorten this sentence (or split into two sentences).

P11 L11: please use a more specific title, so the reader know what this section is about

P11 L14: please better specify that with 'direct organic sedimentation' you mean contributions of litter fall to increases in marsh elevation. The fact that local vegetation has a high biomass production does not necessarily mean that this litter will contribute to long-term accretion rates, so this should be nuanced.

P11 section 4.2.1 In my opinion, the conclusions drawn in this section are too much based on speculations. The only evidence you present that vegetation has an effect on sedimentation rates is that Juncus has a higher standing biomass (while no measures of biomass have been carried out on the studied marsh), without putting forward evidence that e.g. indeed more autochthonous plant material is being retained on the longer term. Moreover, if measurements would have been carried out over e.g. 11 months, the conclusions would have been different and the Sporobolus plots would have collected most sediment. Therefore, I would like the authors to formulate these conclusion more careful and include some discussion about the effect of the duration on their measurements on their results.

P11 L30: please use a more specific title, so the reader knows what this section is about

P11 L34-36: You can add '3) flooding frequency is higher at lower elevations' to this list.

P12 L28 – P13 L5: Here you compare the results from the short-term methods with the long-term methods in order to draw conclusion about the redistribution of surface materials. However, the data obtained with the short-term methods has only been collected in December and January, neglecting potential intra-annual variability in the composition of deposits. This is a major concern of mine, as I don't agree the results obtained in these two months can be directly compared to the results obtained over a 19 months period without addressing this issue thoroughly: please do this.

P13 L28: How about the effect of kinetic fractionation of stable carbon isotopes on the results of your analysis.

P13 L38: In my opinion, I don't agree that the evidence presented allows to draw definite conclusions about the mobilisation of litter on the tidal marsh. Therefore I propose that these results are formulated in terms of hypothesis instead of conclusions.

P13 L40 – P14 L1: 'Autochthonous sedimentation' is a strange term, as sedimentation refers to sediment deposition. This could be changed with 'autochthonous litter'

P14 L5: Please change to e.g. 'Implications for wetland functioning'

P14 L10: Since you didn't measure long-term C sequestration, remove 'sequestration' from the title of this section

P14 L12-14: Since C deposition was only measured 4 events in December and January, I don't agree to calculate annual C deposition rates based on this data, as this way 1) you ignore intra-annual variations in C deposition and 2) the reader might think that you measured C deposition over a whole year. Also, you don't discuss the effectiveness of the method you use to calculate these number (filters) in trapping deposits. I suggest the annual C deposition rates are removed, or a detailed discussion on the effect of intra-annual C deposition dynamics on the calculations is included.

Moreover, you use the results from the filters to calculate these annual C deposition rates, while the amount of deposits measured with the filters (fig. S2) are an order of magnitude smaller compared to the amount of deposits measured with the vials (fig. S3). Please explain why you used the filter results to make these calculations, and not the vial results?

As one of the goals of your study is to compare both the filter and vial method, please provide a more in-depth discussion of the effect of the order of magnitude difference between the results from both methods on the calculations you make and the conclusion you draw based on this data.

P14 L26: By using the title 'Decomposition of organic matter…' you suggest that you have effectively measured OM decomposition, which is not the case. Please change the title so that this is more clear. E.g. 'Chemical structure of deposits varies among…'

P14 L28-31: Please reformulate this sentence: by saying '… these analyses have revealed insights in to fate of aboveground OM and the likelihood of their contribution to…' you suggest that you have done measurements that directly allow you to say something about the different contributions of OM in these different vegetation assemblages to long-term C sequestration. This is however not the case, as you use chemical measurements to make suggestions about these processes.

P15 L8: Based on which data do you calculate the 'retention of plant-derived C'? Please explain.

P15 L17: 'The selective sorption of N by a plant…': how does this explain that Juncus litter is enriched in N compared to the original biomass?

P15 L23: How does table 2 show that the bacterial biomass increases for Sacocornia and Sporobolus?

P15 L24-27: This seems highly speculative and you don't use any data or references to prove this: I suggest you remove this.

P15 L30-31: you only measured C deposition on a very short timescale (averaged over 2 months), so I would refrain from any suggestions or conclusion of your observations for long-term C sequestration.

*Technical corrections*

P1 L15: remove 'surface'

P1 L21: Replace 'Accretion was…' by 'Accretions rates were…'

P1 L23: change '(6d)' to '(6 days)'

P1 L28: change 'mid infrared' to 'mid-infrared' (also in the rest of the ms)

P2 L5: change 'broad' to 'general'

P2 L8: change 'exceptional productivity' to 'exceptionally high productivity'

P2 L12: Change 'Surface elevation and sedimentation dynamics are central…' to 'Sedimentation dynamics partially determine the survival of coastal wetlands under rising…'

P2 L14-16: This is a strange sentence: first you define minerogenic as 'dominated by mineral inputs', by which you imply that there is also other (organic) material present. Next you say that most saltmarsh sediments contain both organic and mineral fractions, repeating what you first said. You can simply only say that most saltmarsh sediments are a mixture of organic and mineral materials, to avoid confusion.

P2 L18: change 'sediment' to 'sediments'

P2 L19-20: change '…); as well as the tidal range of a site and position…' to '…), the tidal range of a site and the position…'

P2 L25: change 'Broadly' to 'Generally'

P2 L26: change 'helping to trap mineral sediments' into 'facilitating sediment tapping'

P2 L27-30: Change to: 'Findings of comparative studies of the effect of vegetation composition on sediment deposition rates, however, vary from no difference among different vegetation species () to substantial differences among…'

P2 L32: I would change this sentence to: 'Average global rates of carbon accumulation in saltmarshes are extremely high, relative to…'

P2 L33: state that SE is the standard error

P2 L39: change 'their' to 'its'

P3 L1: change 'soil pools' to 'soils'

P3 L9: You can change this sentence to 'Because methods vary…, a combination of …'

P3 L15: change 'presented' to 'presents'

P3 L15-16: I would reformulate this sentence and state that another aim of your study was to compare different methods that are used to measure sedimentation rates on tidal marshes (otherwise it is not clear to the reader whether or not you made the comparison).

P3 L24-25: put '(Fig. 1)' at the end of the sentence

P3 L 25-26: 'mangrove species Avicennia…'

P3 L27: 'the upslope limit of saltmarshes…'

P3 L28: 'but for the most part saltmarshes are bordered…'

P3 L29-30: '…  with ranges in elevation and tidal extent.'

P3 L31: 'Salmarshes within this site comprise…'

P3 L31: '… communities.  The lower and middle marsh is characterized by an association of … pathway). The upper marsh …'

P3 L36: 'Fifteen plots were selected on the basis…'

P4 L5: is this g dry weight per $m^{-2}$? If so, mention this, also in the next sentence.

P4 L6: '… 350 g $m^{-2}$). Moreover, there do not'

P4 L12-15: Move these sentence to the beginning of the study area section: they provide general information about SE Australian saltmarshes.

P5 L5: Change 'sedimentation traps' to 'sediment traps'

P5 L35: Change to '… the supernatant decanted and the vial was placed…'

P7 L14-16:  Please explain the symbols more clearly: e.g. 'where $\delta^{13}C$ denotes the isotopic signal of different sources of OC: $C_{residue}$ (…), $C_{C4}$ (…) and $C_{C3}$ (…).

P8 L16: please mention the units of '100 ± 32.73'

P8 L19: better to give the range in $R^2$ instead of saying '$R^2 > 0.35$); I wouldn't call these relationships significant as long as you didn't test them statistically.

P10 L 15-16: change to '… and deposition measured with short-term sediment traps…'

Section 4.1: use the re-calculated accretion rates (see my comments above)

P11 L14: change 'massive' to 'large'

P12 L7: 'the physical position'

P14 L6: This sentence is not correct: change to e.g. '… surface dynamic is critical to predict the survival…'

P14 L11: Please rephrase 'organogenic and minerogenic assemblages' to e.g. 'organogenic and minerogenic deposits'

P14 L28: Replace 'MIR' by 'MIR spectroscopy'

P15 L38: remove 'then'

*Figures*

Figure 1

- Heading: change '…location of nearest…' in '…location of *the* nearest …'

Figure 2

- Heading: is 'SE' the standard error? Is this the same as standard deviation? Please clarify.
- Change the axes so that the 0 marker of the y-axis is at the same height of the x-axis (since you don't plot negative accretion)
- You should make it more clear that what you show is the height of deposited sediments above the marker horizon. Now the reader can interpret it as accretion rates measured at different time periods. I would change the y axis label to something like 'Height of deposited sediments (mm)'

Figure 3

- Heading: write '6d' as '6 days'
- As you have standard deviations on this data the quality of the figure would improve if the differences between the different vegetation species are significantly different, e.g. with letters above the bars.

Figure 4

- The letters written within the symbols of A) are very difficult to read: place them next to the symbols
- Also the letters next to the symbols in A) are difficult to read: enlarge them and increase the space between the symbol and the letters

Figure S1

- Heading: replace 'scatterplots' with 'plots'; explain what 'AHD' is; put 'regression line' in plural; explain that DW (on the y-axis) means dry weight; explain what 'bulk material' is.
- Y-axis: change units to 'g DW m$^{-2}$'
- Plot D should be January 'spring' instead of 'neap'?

Figure S2

- Heading: same remarks as for fig. S1
- Replace the y-axis label as for fig. S1
- Remove 'no linear fits' from the legend: this is already explained in the heading
- Plot D should be January 'spring' instead of 'neap'?

**Tables**

Table 1

- Heading: change 'Summary of sediment measure techniques…' to 'Summary of sedimentation measurement techniques'; Change 'C' to 'OC', since you measure only organic carbon
- Under Parameter, change 'Measure' into 'Measurement'
- Under 'Filter + isotopic analyses': clarify what 'sediment residue' is. This should be clear to the reader without reading the whole manuscript.
- Under 'Filter + MIR & $^{13}$C NMR': change 'Character of …' to 'Characteristics of …'
- In the 'Filter + elemental analysis' section: C deposition rate is expressed in 'yr$^{-1}$' while you only measured for a short period in summer. This should be changed (see my previous comments)
- In the notes ([a]): change '%C' to '%OC', since you measured organic carbon
- For the filter method – 'Filter + isotopic analysis': it should be clear what 'sediment residue' is, please clarify in the heading.

Table 2

- Heading: change 'assemblage' to plural; change '… plant assemblages, plus other…' to 'plant assemblages and other potential sources'; change '… for each of biomass…' to '…for each of *the* biomass…'
- Explain what 'n/a' stands for in the heading

Table S1

- Place 'Number of tides exceeding mean plot elevation' above the names of the neap and spring events to increase readability

Table S2

- Are these values based on 1 measurement or are these average values from multiple replicates? If so, provide the standard deviation.

---

## Author Comment (AC1) · 28 Apr 2017

We thank the two anonymous reviewers for their constructive comments and suggestions. In the pages below we respond to each of these in turn. Unless stated otherwise, we intend to incorporate these changes into a final version of the manuscript.

**REFEREE #1**

REFEREE COMMENT:

Kelleway et al present an interesting work on medium-term and short-term accretion and deposition dynamics in different vegetation communities of a salt-marsh site on the Australian East coast. By combining different methods for measuring short- and medium-term deposition and accretion, they were able to reveal that considerable differences exist between communities with regard to accretion and organic-matter source. The manuscript presents some novel aspects on sediment and organic matter dynamics within salt-marsh systems. Unfortunately, however, I cannot recommend the work for publication before several shortcomings, often with regard to the structure of the ms, have been considered. Overall, the connection between hypotheses/research questions and the rest of the ms is very weak. Thus, large parts of the discussion have not been sufficiently set up in the introduction and particularly not in the hypotheses. No doubt the study used interesting methodology and a wide array of tools; however, in most parts it does not become clear to the reader why certain analyses/methods were conducted or why they are necessary at all until one gets to the respective parts in the discussion of the ms. The authors need to make clear that this work is not simply about comparing different methods for assessing deposition, accumulation, and accretion dynamics. I will try to elaborate on this in the following:

RESPONSE: We are willing to make the necessary changes to manuscript structure in order to clarify hypotheses and to link these to both methodology descriptions and discussion points. We have provided more detailed responses below regarding the specific comments raised by the referee.

REFEREE COMMENT: Title page: L1 The title could be more specific, but I don't have strong opinions on that. It seems that throughout the ms you rather use the terms deposition and accretion. So why is "accumulation" used in the title?

RESPONSE: We intend to modify the title to "Sediment and carbon deposition vary among vegetation assemblages in a coastal saltmarsh" in line with this comment and a similar comment from referee 2.

REFEREE COMMENT: L18 Please make clear that this is a case study, conducted in one marsh system only. "within 3 vegetation types common throughout Australia" could be misleading and can give the impression that this is a larger scale study which has been replicated in several systems. Please, also discuss implications of this missing replication.

RESPONSE: Text will be changed to improve clarity. Discussion of the implications and limitations of using a single study site will be added, in addition to text already included regarding the need for further research across a broader range of geographic settings (P16:L25-30)

REFEREE COMMENT: Main part: L1+3 please be consistent in your wording coastal wetland <-> coastal saltmarsh, please use different terms only if you mean different things, otherwise that can be confusing..

RESPONSE: Terminology will be updated to ensure consistency and avoid confusion.

REFEREE COMMENT: L13 give correct reference Kirwan instead if "Kirwin"

RESPONSE: Spelling will be updated

REFEREE COMMENT: L18 you use the term sediment for both, the suspended matter that can deposit on the marsh surface but also to that what others refer to the "soil" of the marsh. I know, that is hair-splitting, but please make sure that you don't confuse the reader too much. Especially when you are talking about organogenic systems (L15), you should not use the term sediment when actually referring to something like a peat soil. Please check out "Do marine rooted plants grow in sediment or soil? A critical appraisal on definitions, methodology and communication" (Kristensen and Rabenhorst 2015) for clarity.

RESPONSE: Thank you for pointing this out. Manuscript will be updated to use both terms as relevant.

REFEREE COMMENT: L24-25 There is also work focused on other species (Schoenoplectus) by for instance Langley or Langley and Megonigal (PNAS or Nature) or by Rooth (2003) on Phragmites that could be mentioned here.

RESPONSE: Reference to these other species and relevant studies will be added

REFEREE COMMENT: L39 I think this study of Kirwan et al (2013) was only on decay but not on the balance between OM inputs and decay. I think Mueller et al (2016; GCB) is more focused on the link between the two or Kirwan and Megonigal (2013; Nature) at least discusses both.

RESPONSE: Reference will be updated to Mueller et al (2016; GCB).

REFEREE COMMENT: Page 3 L2 Hemminga and Buth 1991 give a nice citation here on litter-quality effects on decay

RESPONSE: Reference to this paper will be made, noting that they found chemical composition of the plant material to have an impact upon decomposition rates of halophyte litter.

REFEREE COMMENT: Page3 L13-15 Please give expected directions of effects in your hypotheses instead of only expecting that they will "vary".

RESPONSE: Text can be changed to: "We hypothesise that: 1) mineral deposition and accretion will be highest in lower elevation assemblages but organic deposition and accretion will be highest in the

*Juncus* assemblage; and 2) the source and character of material deposited will vary temporally according to tidal inundation patterns, with a greater proportion of allochthonous material deposited during times of high inundation frequency."

REFEREE COMMENT: L15-18 It seems like the second aim of this study is a methods comparison. I see this as a major weakness of the manuscript. Like mentioned above, either justify why the application of the different methods was necessary to answer you research questions or save that for a very nice second manuscript. Otherwise it is hard to follow your structure.

RESPONSE: This experiment was not set up as a formal comparison of different methods. We acknowledge that the current wording of the manuscript may imply that, however, as both referees have raised this issue. In a revised manuscript, we hope to outline the rationale for using three different, but complimentary methods, and what insights we gain from the methods used. We believe this has largely been done in the manuscript already, but can be clarified and expanded upon, while also improving the manuscript structure.

REFEREE COMMENT: Page4 L4,5 give range or st deviation for biomass values

RESPONSE: Range values will be added to the existing text:

"*Juncus* mean = 1116 g m$^{-2}$, range = 51-4832 g m$^{-2}$), compared to that of the non-rush assemblages (*Sarcocornia* mean = 317 g m$^{-2}$, range = 52-1184 g m$^{-2}$; *Sporobolus* mean = 349 g m$^{-2}$, range = 148-852 g m$^{-2}$)"

REFEREE COMMENT: L11 to what depths was biomass assessed here?

RESPONSE: Clarke and Jacoby 1994 report belowground biomass from the 0-20 cm depth interval. This information will be added here.

REFEREE COMMENT: L23+L32 Briefly mention why those measurements were conducted and don't just list them. Well, an informed reader can probably guess why you measured elevation or deployed marker horizons; however, when it comes to 2.5(isotopes) or 2.7 (13CNMR) you need to give a rationale.

RESPONSE: The following sentences will be added:

P4, L23: "Elevation was recorded to assess relationships between deposition dynamics and plot position within the tidal frame."

P4, L32: "The feldspar marker horizon (MH) technique was used to record the amount of accretion of bulk materials at each plot"

P6,L5: "Elemental C and N content was measured in order to quantify C deposition rates and infer biomass, litter and soil consumption 'quality' (C:N). $\delta^{13}$C was analysed to infer the source of samples relative to reference sources material and literature values."

P6,L20: "Solid-state $^{13}$C nuclear magnetic resonance (NMR) spectroscopy was used to quantify the contribution of C functional groups to live plant biomass, litter and residue samples. This was carried out to identify what compositional changes occurred between the different sample types, and to what extent this differed between vegetation assemblages and inundation periods."

REFEREE COMMENT: Page6 L21 which functional groups, why was this done?

RESPONSE: These functional groups (and their spectral regions) are detailed later in the paragraph:

"organic functional groupings found in natural organic materials: Alkyl C (0-45 ppm), N-Alkyl/Methoxyl (45-60 ppm), OAlkyl (60-95 ppm), Di-O-Alkyl (95-110 ppm), Aryl (110-145 ppm), O-Aryl (145-165 ppm), Amide/Carboxyl (165-190 ppm) and Ketone (190-215 ppm)."

This was carried out to identify what compositional changes occurred between the different sample types, and to what extent this differed between vegetation assemblages and inundation periods (see response above).

REFEREE COMMENT: Page7 L17 Is this method needed to better interpret isotope data?

RESPONSE: No, this method is not directly related to the interpretation of isotope data. It is instead used to inform differences in the molecular composition of samples and is used in interpreted in concert with $^{13}$C NMR (the latter gives more detailed information regarding composition but was limited in its use due to practical constraints)

REFEREE COMMENT: L24 did you really assess net accretion or accumulation?

RESPONSE: 'Net accretion (i.e. vertical surface accumulation)' was recorded across the 19 month period for the *Juncus* assemblage. Wording of this sentence will be changed to clarify this.

REFEREE COMMENT: Page10 L32 why are you using "organogenic" instead of "organic" deposition?

RESPONSE: Term will be replaced with "organic"

REFEREE COMMENT: Page12 L28-34 I think this is a real highlight of your study. Try to better set up this whole redistribution thing in your intro. I guess there is relatively little known about these dynamics.

RESPONSE: We thank the reviewer for this suggestion. We intend to update the introduction to better highlight this as a focus of the study.

REFEREE COMMENT: Page13 L16-18 I don't buy that based on 13C natural abundance only! Did you consider that 13C-fractionation processes during of organic matter decay are inducing shifts in your signatures? Are differences between litter and fresh biomass large in your species? Can your other methods support/help here?

RESPONSE: Our data presented in Table 2 show that differences in δ¹³C between fresh biomass and partially decomposed litter samples are small. That is, mean values are within 1‰ of one another, and δ¹³C variability among replicate samples is typically low. Further, there is not a consistent direction of fractionation among the three species analysed (i.e. litter is less negative than biomass for *Sarcocornia*, but more negative for *Sporobolus* and *Juncus*).

In addition to the above, we note that a number of other studies have shown that there is little to no difference in δ¹³C between fresh and decomposing leaves of estuarine plant species (e.g. Zieman et al., 1984;Fry and Ewel, 2003;Saintilan et al., 2013), though the literature record is very limited in terms of species analysed.

We cannot rule out the potential for ¹³C-fractionation occurring in the decay from litter to residue samples. Unfortunately, no controlled experiments have been undertaken to assess this. We intend to highlight this as an uncertainty in our method and suggest the need for further research in this regard. While we recognise the uncertainty associated with isotope fractionation, the isotope method was just one of three lines of evidence used to support our conclusion of redistribution of surface materials (see section 4.2.3).

REFEREE COMMENT: Page14 L11 following: I think it goes too far to discuss sequestration rates based on the presented data. You studied processes on the marsh surface, which may affect C sequestration, but here you should really stick to "deposition". Also "surface C sequestration" sounds odd to me. I don't know if C sequestration can be determined at the surface if a more or less permanent process is meant. It needs to become clear that deposition, accumulation, and sequestration are different processes. Further down in the paragraph you are using accumulation again. Please be sure to be consistent in the use of terminology.

RESPONSE: The referee raises a good point here. We intend to clarify terminology here, modify the discussion to focus upon short- to medium-term patterns of deposition and accumulation (for which we have data) and limit discussion of 'sequestration' to literature that consider carbon sequestration processes over longer time frames. We also intend to change the title of the manuscript to refer to the focus on 'deposition' patterns rather than accumulation or sequestration patterns.

REFEREE COMMENT: L26 and the whole paragraph: You don't have a hypothesis on decomposition. This needs to be linked!

RESPONSE: Thank you for raising this omission. We intend to add the following hypothesis to the introduction text:

"3) we hypothesise that there will be no difference in biomass-litter-sediment decay patterns among the vegetation assemblages".

We intend to update the relevant discussion text to link directly to this hypothesis.

REFEREE COMMENT: Page15 L21 "Reddy and DeLaune 2008" is a nice textbook indeed, but I know there is a bunch of peer-reviewed primary research or even review articles out there that should be rather cited here!

RESPONSE: We propose to replace this citation, with the following:

Sterner, R. W. and Hessen, D. O.: Algal nutrient limitation and the nutrition of aquatic herbivores, Annual review of ecology and systematics, 25, 1-29, 1994.

Hessen, D. O., Elser, J. J., Sterner, R. W., and Urabe, J.: Ecological stoichiometry: An elementary approach using basic principles, Limnology and Oceanography, 58, 2219-2236, 2013.

**REFEREE #2**

REFEREE COMMENT: Kelleway et al. present a study on the effect of different vegetation species on the trapping of mineral and organic deposits on a tidal marsh in southeast Australia. They use three different methods to assess deposition rates at the short (days) and medium term (months). Their study provides insights in the processes controlling both mineral sediment deposition and deposition of organic matter on a tidal marsh platform. Although their results make a substantial contribution to our knowledge of processes controlling tidal marsh growth and organic carbon dynamics in these environments, some issues need to be resolved before publication of the manuscript is possible, as I point out in my comments below.

General comments

REFEREE COMMENT: One of my main concerns is that the authors use short-term (days) deposition data measured only in December and January to draw conclusion on longer term carbon and sediment dynamics, since they express the accumulation rates on a per-year basis. I think the authors should limit the conclusions they draw based on these data to short-term deposition rates, instead of C sequestration.

RESPONSE: The referee raises a good point here. We intend to clarify terminology and focus upon short- to mid-term patterns of deposition and accumulation (for which we have data) and limit discussion of 'sequestration' to literature that consider carbon sequestration processes over longer time frames. We also intend to change the temporal reporting units and change the title of the manuscript to refer to the focus on 'deposition' patterns rather than sequestration patterns.

REFEREE COMMENT: Another concern is that the authors calculate the annual sediment deposition rates using a linear regression line which does not pass through the origin to: this results in an overestimation which has to be corrected.

RESPONSE: We believe it is more appropriate to use the regression approach in the manuscript (i.e. one without forcing a y intercept of 0) than the approach the referee suggests. A detailed rationale for this is discussed in relation to the specific comment about this by the referee below

REFEREE COMMENT: Furthermore, the authors compare the results of measurements at different timescales (days – months) to draw conclusions about the processes controlling accretion rates in different vegetation assemblages. However, they do not address the issues this poses, e.g. the short-term methods were only employed in December and January, so no information from the rest of the year is collected using these methods. This will have an effect on the results, and should be thoroughly addressed.

RESPONSE: We chose to base our sampling strategy upon expected tidal inundation patterns rather than to capture seasonal variability for several reasons. First, based on relevant literature (Rogers *et al.*, 2013) we expect tidal inundation patterns to be of primary importance to deposition and accretion dynamics. Second, we do not expect there to be substantial seasonal variability due to factors other than tidal pattern variation. That is, the study region does not experience high seasonal variability in rainfall (a point we failed to mention in the methods section, but intend to address), nor are there clear seasonal patterns in terms of biomass standing stock or senescence (Clarke and Jacoby, 1994).

Having said that, we agree that we have not adequately addressed this point in the manuscript. We propose to more clearly state why we expect little seasonal variation in deposition, discuss our study limitations and to apply caution in comparing results between different timescales.

REFEREE COMMENT: Also, as one of the aims of this paper is to compare different methods, I would like a discussion about the effect of the results obtained by these different methods on the conclusions they draw. The filter and vial methods result in C deposition rates that differ up to an order of magnitude: this is now not discussed in the manuscript and is a major shortcoming, and necessary if the authors want to use these results in order to draw conclusion based on these data.

RESPONSE: It was not our intention that this manuscript be seen as a formal methods comparison, though we acknowledge that both referees have taken this impression. As outlined in the introduction and methods sections, the methods chosen vary in their effectiveness of trapping and retaining different materials. For this reason, a combination of techniques was used to infer the relative importance of different physical and biotic influences on deposition and accretion. While the results from each method are informative in their own right, in most cases the results from these methods are not directly comparable (and may be expected to have an order of magnitude difference).

We propose that our sentence "this study also presented an opportunity to compare wetland sedimentation methods" (and any others like it) will be removed from the introduction. We hope that this removes any impression that a formal methods comparison is an aim of this manuscript.

In a revised manuscript, we hope to outline the rationale for using three different, but complimentary methods, and what insights we gain from the methods used. We believe this has largely been done in the manuscript already, but can be clarified and expanded upon, while also improving the manuscript structure.

REFEREE COMMENT: The authors should re-consider the title they use for this manuscript, e.g. based on the work they present, the word 'deposition' could replace the word 'accumulation'.

RESPONSE: We intend to modify the title to "Sediment and carbon deposition vary among vegetation assemblages in a coastal saltmarsh" in line with this comment and a similar comment from referee 1.

Specific comments

REFEREE COMMENT: P1 L24: you report the C deposition rates on a yr-1 basis, while you only measured during 2 cycles of spring and neap tide, which is misleading for the reader. I address this issue further in my comments.

RESPONSE: All deposition rates will be changed to a $d^{-1}$ (day) basis.

REFEREE COMMENT: P1 L33: By stating in the abstract that you have gained novel insights into processes responsible for regional differences, you suggest that you explicitly addressed these issues at a broader regional scale, which is not the case. Furthermore, by saying '…processes responsible

for regional differences…', you suggest that you have conclusive evidence that these processes are the most important one, which is also not the case (e.g. you didn't took belowground biomass production or soil compaction into account). Therefore, I would change this sentence so that you make it clear you only performed these analyses for a single tidal marsh.

RESPONSE: Sentence will be re-written to ensure it is clear this study was conducted in a single marsh. Wording regarding regional differences will be softened to identify the processes we have measured and those we have not.

REFEREE COMMENT: P1 L34-36: I would formulate this more careful, as now you imply that it is possible that belowground processes are of minor importance. This contradicts with the finding of e.g Saintilan et al. (2013) that root OC is an important component of the total OC pool in SE Australian saltmarshes, and P2 L34 of this ms.

RESPONSE: This sentence was not intended to take that meaning. Sentence will be revised to emphasise the fact that belowground processes can be important.

REFEREE COMMENT: P4 L22: I recommend to change this title to e.g. 'Surface elevation measurements'

RESPONSE: Agreed. Will change as suggested.

REFEREE COMMENT: P4 section 2.2: This method is of course characterised by substantial uncertainty. Are there e.g. no measurements of daily tidal height in the surroundings of the study area? That way you could calculate the difference between measured and predicted tidal height and use this to correct your measurements?

RESPONSE: There was an error in the wording here. The measured tidal height (at a nearby gauge shown in Figure 1), rather than the predicted tidal height was used to calculate plot surface elevation. Sentence will be updated to "Depth of inundation above the saltmarsh surface was measured immediately after the tide receded and subtracted from the *measured* tide height to obtain an estimate of surface elevation."

REFEREE COMMENT: P5 L4: I propose you change this title to 'Sediment traps'

RESPONSE: Agreed. Will change as suggested.

REFEREE COMMENT: P5 L5-6: Here you state that the purpose is to quantify short-term deposition, while you report the measurement on a yr-1 basis. This should be addressed (see below).

RESPONSE: All deposition rates will be changed to a $d^{-1}$ (day) basis.

REFEREE COMMENT: P5 L14: please explain what you mean by 'resolution'

RESPONSE: This refers to the smallest accumulation increment detectable by a given method. Text will be updated to describe this.

REFEREE COMMENT: P6 L2: The term 'residual sediment' is confusing, as this term is used to denote both residual sediments and organic matter, I propose to change this to e.g.; 'residual deposits'.

RESPONSE: Agreed. Will change as suggested.

REFEREE COMMENT: P6 L6: Jaschinski et al. (2008) is not included in the reference list

RESPONSE: Reference will be added. Citation is:

Jaschinski, S., Hansen, T., and Sommer, U.: Effects of acidification in multiple stable isotope analyses, Limnology and Oceanography: Methods, 6, 12-15, 2008.

REFEREE COMMENT: P6 L11: It's very confusing that you say here that you used MIR spectroscopy to assess the composition of the listed materials, as this is not done in this ms. In some cases, MIR spectroscopy is used to assess characteristics of the analysed material (e.g. C content), based on a calibration dataset, but this is not done here. I think this sentence is confusing to the reader, as you use the MIR spectroscopy results only to perform a PCA to discriminate between different types of deposits. Therefore, I would limit the materials section about MIR spectroscopy to this aspect.

RESPONSE: We do not agree with the referee's comment. MIR was used in conjunction with $^{13}$C NMR to assess the composition of materials. That is, we use MIR primarily to assess the variability in spectra of all samples analysed in terms of their bulk composition (mineral plus organic components). We believe this variability in spectra is best presented by PCA. We then use the loadings plots of the PCA to assess what materials (e.g. quartz, kaolinite, water and OM-alkyl) are contributing to among sample variation based on diagnostic MIR spectral peaks. This is presented in section 3.4 and Figure 4B and C. We then use $^{13}$C NMR to provide more specific information on the composition of the organic matter present in each of the samples.

We note that discussion of MIR results is limited to data that is already presented in the manuscript (i.e. the PCA and related loading plots in Figure 4). Individual MIR spectra can be presented as part of the supplementary information, if requested.

We believe the detail of MIR methods and rationale should be retained in the manuscript.

REFEREE COMMENT: P6 L14: if these procedures are important for the reader to replicate your measurements, please mention them.

RESPONSE: The central information (instrument, spectral range, resolution) required to replicate the method is already detailed in our manuscript. Reference to the cited paper is included for readers who wish to access further detail.

REFEREE COMMENT: P6 L15: the mid-infrared range of the electromagnetic spectrum is between 4000 – 400 cm$^{-1}$, so why did you measure between 8000 – 400 cm$^{-1}$? Please clarify.

RESPONSE: The infrared spectrometer has an operating range of 8000-400 cm$^{-1}$. All samples were scanned over this entire range.

Although the upper limit of the MIR region is 4000 cm$^{-1}$, we have included the signal between 6000 and 4000 cm$^{-1}$ in our analysis for two reasons.

Firstly, in many samples significant signal intensity existed at 4000 cm$^{-1}$. If we had truncated our spectra to this wavenumber limit, the baseline offset transformation would not have worked correctly as real signal would have been lost differentially from the various samples. By extending our spectra to 6000cm$^{-1}$, a region devoid of signal intensity was present that could be used to appropriately apply the baseline offset spectral transformation uniformly across all spectra.

Secondly, the 6000-4000 cm$^{-1}$ wavenumber region contains the first NIR overtones of the MIR spectra and thus may contain useful information that may aid in the development of predictive models.

The text will be amended to include these justifications.

REFEREE COMMENT: P6 L15: Please clarify why the spectral range was adjusted to 6000 – 600 cm-1?

RESPONSE: The spectral range was limited because at wavenumbers >6000 cm$^{-1}$ and <600 cm$^{-1}$ noise in the acquired signal intensity was evident. At wavenumbers <450 cm$^{-1}$ spikes in observed signal intensity were also evident for some samples. As a result, the spectra were truncated to 6000-600 cm$^{-1}$. The text will be amended to indicate why the spectra were truncated to the 6000-600 cm$^{-1}$ range.

REFEREE COMMENT: P6 L35-36: You say you test main and interactive effects of vegetation assemblages: please provide the effect on what exactly?

RESPONSE: Sentence will be updated to "to test main and interactive effects of vegetation assemblage (*Sarcocornia, Sporobolus, Juncus*) and tidal event (repeated measures: December neap, December spring, January neap, January spring) on the amount of material retained at the end of a deployment period."

REFEREE COMMENT: P6 Section 2.8: please mention that you report the confidence on the mean of replicate measurements as standard error (as I assume this is what you mean with SE). Also state how this was done an why you didn't use standard deviation to report on the spread among different replicate measurements.

RESPONSE: We chose to report the standard error as this incorporates the number of samples contributing to the mean and its confidence. As mentioned elsewhere in the methods, for some measures a small number of samples were excluded from analyses – therefore we report standard error. We intend to add this detail to Section 2.8.

REFEREE COMMENT: P6 Section 2.8 + P7 L24-29 + section 4.1: You use a simple linear regression, which you fit through the data points representing sedimentation rates above the MH's, to obtain annual rates of sediment deposition: this technique leads to an overestimation of sediment deposition rates! As you show in figure 2: the regression lines do no pass through the origin of the graph, which implies that after an infinitesimal timestep you have e.g. already 0.5mm accretion at the Sporobulus site. Likewise, when you use this regression line to calculate the amount of material that has been deposited after 12 months, you will overestimate this amount. This should be corrected: make sure you force your regression line to pass through the origin and calculate the deposition rates again.

RESPONSE: We believe it is more appropriate to use the regression approach in the manuscript (i.e. one without forcing a y intercept of 0) than the approach the referee suggests. While there would seem to be a logical argument for forcing the regression to pass through the origin in relation to the marker horizon (i.e. at time = 0 there was no accumulation above the marker horizon), forcing the intercept places undue importance on a nil accretion value at time = 0. This is misleading in terms of what is happening on the marsh surface at the time of marker horizon deployment - in reality there would have been some, unquantified accretion or erosion occurring at this time point.

The approach that we use is in fact the one which is more conservative overall and less likely to overestimate accretion dynamics. This is true of both the rates of accumulation and the strength of the linear relationships. To demonstrate this, we have tabulated the results of linear regression analyses using both our method (not forcing y-intercept = 0), and that suggested by the referee (forcing y-intercept = 0), here:

| | Approach used: Not forcing y-intercept = 0 | | Forcing y-intercept = 0 | |
|---|---|---|---|---|
| Vegetation assemblage | Linear accretion rate (mm) ± SE | $R^2$; P-value | Linear accretion rate (mm y$^{-1}$) ± SE | $R^2$; P-value |
| *Sarcocornia* | 0.78 ± 0.18 | $R^2$= 0.16; P<0.001 | 0.92 ± 0.09 | $R^2$= 0.59; P<0.001 |
| *Sporobolus* | 0.88 ± 0.22 | $R^2$= 0.14; P<0.001 | 1.30 ± 0.11 | $R^2$= 0.65; P<0.001 |
| *Juncus* | 1.74 ± 0.13 | $R^2$= 0.68; P<0.001 | 1.70 ± 0.06 | $R^2$= 0.91; P<0.001 |

REFEREE COMMENT: P7 L9: Please explain what you mean by 'organic residue': are these the deposited macrolitter? Or all deposited materials combined? Or…?

RESPONSE: The 'organic residue' is inclusive of all the organic material which was leftover after macrolitter was removed. It is the material that could not be visually identified and accounted for in the physical sorting procedure. We intend to clarify this definition in an updated manuscript.

REFEREE COMMENT: P7 L18-19: please explain what you mean with 'composition'? It's confusing that you state that you will identify differences in composition, while you will only use PCA to plot the data on two PC's. Please better explain here how you used the PCA based on MIR spectra as an added value to standard lab analyses.

RESPONSE: As stated in a response above, we use MIR primarily to assess the variability in spectra of all samples analysed in terms of their bulk composition (mineral plus organic components). We believe this variability in spectra is best presented by PCA. We then use the loadings plots of the PCA

to assess what materials (e.g. quartz, kaolinite, water and OM-alkyl) are contributing to among sample variation based on diagnostic MIR spectral peaks.

REFEREE COMMENT: P7 L24: I don't agree that 'consistent' accretion was measured for the Juncus plots, as e.g. replica 2 remains relatively stable after 11 months and replica 1 and 3 show negative erosion rates towards the end of the measurement period. I would formulate this more careful.

RESPONSE: Agreed. We intend to revise our wording here.

REFEREE COMMENT: P8 L12: F2,45.8: how can the degrees of freedom of variance within groups be 45.8?

RESPONSE: Non-integer degrees of freedom can occur in mixed models and are common in repeated measures analysis. This is because the denominator (or within groups) degrees of freedom are calculated based on the model and the estimated random effect and repeated measure matrices.

REFEREE COMMENT: P8 L24-25: please perform a statistic to show whether the differences between sporobolus and the other vegetation types is significant.

RESPONSE: We intend to update the text to include the pairwise comparison result for this comparison (not reported previously):

"Bulk deposition on filters varied among vegetation assemblages ($F_2, 30.85 = 48.82$; $P = 0.004$), with significantly lower deposition in *Sporobolus* plots relative to both *Sarcocornia* (Bonferroni adjusted P-value = 0.010) and *Juncus* (Bonferroni adjusted P-value = 0.023) plots across all tidal events (Fig. 3; Table 1)."

REFEREE COMMENT: P8 L 31: indicate if the 66% and 78% are mass percentages or some other measure?

RESPONSE: Yes, these are mass percentages. Text will be updated to reflect this.

REFEREE COMMENT: P8 L32-37: Sarcocornia and Sporobolus plots were located on the low marsh, which are generally subject to higher water flow velocities compared to the high Juncus marsh. This can partly contribute to the lower amount of litter retained at the low marsh. Please discuss this briefly.

RESPONSE: We intend to add a brief discussion of this point to section 4.2.3, which already discusses expected differences in hydrodynamic energy across the marsh elevation profile.

REFEREE COMMENT: P10 L9: Indicate that you analysed the types of materials deposited for the short term

RESPONSE: Text will be updated as per comment.

REFEREE COMMENT: P10 L22: how about the effect of sediment removal through erosion?

RESPONSE: Erosion will be added as a potential cause of this variability.

REFEREE COMMENT: Section 4.1: here you discuss that sedimentation rates are higher for the high marsh compared to the low marsh, which is the opposite of what is normally observed. Discuss this briefly, or refer to where you discuss this (section 4.2.2)

RESPONSE: We intend to include reference to section 4.2.2, where this is discussed in detail.

REFEREE COMMENT: P10 L29: Section 4.2 has a confusing structure: in sections 4.2, 4.2.2 and 4.2.3 you describe the results from the short-term methods, while in section 4.2.1 you describe results from the long-term methods. Please indicate this e.g. in the titles of the different subsections, as this in very confusing for the reader.

RESPONSE: The intent of section 4.2 is to discuss both the short-term and medium-term results as indicated by both the first and last sentences of the introductory paragraph of Section 4.2 (i.e. P10,L30-31 and P11,L8-10). For example, Section 4.2.1 describes results from both short-term and medium-term methods, and infers that results from the short-term measures may partly explain the results obtained from medium-term measures.

We intend to use terminology such as 'short-term filter', 'short-term vial' and 'medium-term marker horizon' (as opposed to just 'filter', 'vial', 'marker horizon') throughout this section to clarify the temporal resolution of results being discussed.

REFEREE COMMENT: P11 L6: you state here that during the January neap there was no inundation of the Sporobolus plots for the vials, but in table 2 you report deposition rates for JN in vials for Sporobolus plots. Please explain.

RESPONSE: In the neap tide periods, some deposition and retention of materials was recorded during periods where no tidal inundation is expected to have occurred (based on plot elevations and nearby tidal height measurements). In these instances, non-tidal processes such as rain- or wind-driven sedimentation and/or bioturbation are the most likely causes. Although filters with visible crab-excavated sediment (n = 23/180) were excluded from analysis, such clear identification was not able to be determined for sediments deposited in vials.

REFEREE COMMENT: P11 L8-10: please better explain which 'scale differences' you mean and shorten this sentence (or split into two sentences).

RESPONSE: We intend to re-write this sentence as:

"In the following sections we interpret the influence of biological, physical and interactive processes on saltmarsh surface dynamics. We do so by assessing the response of different surface deposition measures (2 short-term; 1 medium term) among the three vegetation assemblages studied. "

REFEREE COMMENT: P11 L11: please use a more specific title, so the reader know what this section is about

RESPONSE: We intend to replace this with "The influence of vegetation on saltmarsh surface deposition"

REFEREE COMMENT: P11 L14: please better specify that with 'direct organic sedimentation' you mean contributions of litter fall to increases in marsh elevation. The fact that local vegetation has a high biomass production does not necessarily mean that this litter will contribute to long-term accretion rates, so this should be nuanced.

RESPONSE: Yes, we are referring to litter fall here and intend to update the sentence to clarify this. We accept that biomass production does not necessarily equate to higher accretion rates. We also discuss, however, our results of relatively high litter retention in the *Juncus* assemblage relative to other assemblages (Section 4.2.3), while our spectrometric techniques revealed a high contribution of plant-derived C to benthos in the *Juncus* assemblage (Section 4.3.2)

REFEREE COMMENT: P11 section 4.2.1 In my opinion, the conclusions drawn in this section are too much based on speculations. The only evidence you present that vegetation has an effect on sedimentation rates is that Juncus has a higher standing biomass (while no measures of biomass have been carried out on the studied marsh), without putting forward evidence that e.g. indeed more autochthonous plant material is being retained on the longer term. Moreover, if measurements would have been carried out over e.g. 11 months, the conclusions would have been different and the Sporobolus plots would have collected most sediment. Therefore, I would like the authors to formulate these conclusion more careful and include some discussion about the effect of the duration on their measurements on their results.

RESPONSE:

We intend to update this section with discussion of:

- variation among the three vegetation assemblages in terms of the contribution of autochthonous litter to short-term deposition (Figure 3), corresponding to literature (and visually observed) biomass patterns and plant structural differences;

- reference to section 4.3.2 which details variation in the contribution of plant-derived C to short-term benthos among the three vegetation assemblages, as revealed by our spectrometric methods;

- discussion of limitations of our approach for determining the long-term contribution of plants to marsh accretion (including discussion of the effect of measurement duration in our study).

REFEREE COMMENT: P11 L30: please use a more specific title, so the reader knows what this section is about

RESPONSE: We intend to replace this with "The influence of physical factors on saltmarsh surface deposition"

REFEREE COMMENT: P11 L34-36: You can add '3) flooding frequency is higher at lower elevations' to this list.

RESPONSE: Agreed. Thank you.

REFEREE COMMENT: P12 L28 – P13 L5: Here you compare the results from the short-term methods with the long-term methods in order to draw conclusion about the redistribution of surface materials. However, the data obtained with the short-term methods has only been collected in December and January, neglecting potential intra-annual variability in the composition of deposits. This is a major concern of mine, as I don't agree the results obtained in these two months can be directly compared to the results obtained over a 19 months period without addressing this issue thoroughly: please do this.

RESPONSE: We chose to base our sampling strategy upon expected tidal inundation patterns rather than to capture seasonal variability for several reasons. First, based on relevant literature (Rogers *et al.*, 2013) we expect tidal inundation patterns to be of primary importance to deposition and accretion dynamics. Second, we do not expect there to be substantial seasonal variability due to factors other than tidal pattern variation. That is, the study region does not experience high seasonal variability in rainfall (a point we failed to mention in the methods section, but intend to address), nor are there clear seasonal patterns in terms of biomass standing stock or senescence (Clarke and Jacoby, 1994).

Having said that, we agree that we have not adequately addressed this point in the manuscript. We propose to more clearly state why we expect little seasonal variation in deposition, and to apply caution in comparing results between different timescales.

In addition, we intend to use terminology such as 'short-term filter', 'short-term vial' and 'medium-term marker horizon' (as opposed to just 'filter', 'vial', 'marker horizon') throughout the manuscript to clarify the temporal resolution of methods being discussed. Also, all deposition rates will be changed to a $d^{-1}$ (day) basis.

REFEREE COMMENT: P13 L28: How about the effect of kinetic fractionation of stable carbon isotopes on the results of your analysis.

RESPONSE: Our data presented in Table 2 show that differences in $\delta^{13}C$ between fresh biomass and partially decomposed litter samples are small. That is, mean values are within 1‰ of one another, and $\delta^{13}C$ variability among replicate samples is typically low. Further, there is not a consistent direction of fractionation among the three species analysed (i.e. litter is less negative than biomass for *Sarcocornia*, but more negative for *Sporobolus* and *Juncus*).

In addition to the above, we note that a number of other studies have shown that there is little to no difference in $\delta^{13}C$ between fresh and decomposing leaves of estuarine plant species (e.g. Zieman et al., 1984;Fry and Ewel, 2003;Saintilan et al., 2013), though the literature record is very limited in terms of species analysed.

We cannot rule out the potential for $^{13}C$-fractionation occurring in the decay from litter to residue samples. Unfortunately, no controlled experiments have been undertaken to assess this. We intend

to highlight this as an uncertainty in our method and suggest the need for further research in this regard. While we recognise the uncertainty associated with isotope fractionation, the isotope method was just one of three lines of evidence used to support our conclusion of substantial redistribution of surface materials (see section 4.2.3).

REFEREE COMMENT: P13 L38: In my opinion, I don't agree that the evidence presented allows to draw definite conclusions about the mobilisation of litter on the tidal marsh. Therefore I propose that these results are formulated in terms of hypothesis instead of conclusions.

RESPONSE: We propose to review this section to replace reference to 'conclusions' with 'hypotheses'

REFEREE COMMENT: P13 L40 – P14 L1: 'Autochthonous sedimentation' is a strange term, as sedimentation refers to sediment deposition. This could be changed with 'autochthonous litter'

RESPONSE: We propose to change this simply to 'autochthonous materials'

REFEREE COMMENT: P14 L5: Please change to e.g. 'Implications for wetland functioning'

RESPONSE: Agreed

REFEREE COMMENT: P14 L10: Since you didn't measure long-term C sequestration, remove 'sequestration' from the title of this section

RESPONSE: The referee raises a good point here. We intend to clarify terminology here, modify the discussion to focus upon short- to medium-term patterns of deposition and accumulation (for which we have data) and limit discussion of 'sequestration' to literature that consider carbon sequestration processes over longer time frames. We also intend to change the title of the manuscript to refer to the focus on 'deposition' patterns rather than accumulation or sequestration patterns.

REFEREE COMMENT: P14 L12-14: Since C deposition was only measured 4 events in December and January, I don't agree to calculate annual C deposition rates based on this data, as this way 1) you ignore intra-annual variations in C deposition and 2) the reader might think that you measured C deposition over a whole year. Also, you don't discuss the effectiveness of the method you use to calculate these number (filters) in trapping deposits. I suggest the annual C deposition rates are removed, or a detailed discussion on the effect of intra-annual C deposition dynamics on the calculations is included.

RESPONSE: We chose to base our sampling strategy upon expected tidal inundation patterns rather than to capture seasonal variability for several reasons. First, based on relevant literature (Rogers *et al.*, 2013) we expect tidal inundation patterns to be of primary importance to deposition and accretion dynamics. Second, we do not expect there to be substantial seasonal variability due to factors other than tidal pattern variation. That is, the study region does not experience high seasonal variability in rainfall (a point we failed to mention in the methods section, but intend to address), nor

are there clear seasonal patterns in terms of biomass standing stock or senescence (Clarke and Jacoby, 1994).

Having said that, we agree that we have not adequately addressed this point in the manuscript. We propose to more clearly state why we expect little seasonal variation in deposition, and to apply caution in comparing results between different timescales.

In addition, all deposition rates will be changed to a $d^{-1}$ (day) basis.

REFEREE COMMENT: Moreover, you use the results from the filters to calculate these annual C deposition rates, while the amount of deposits measured with the filters (fig. S2) are an order of magnitude smaller compared to the amount of deposits measured with the vials (fig. S3). Please explain why you used the filter results to make these calculations, and not the vial results?

RESPONSE: As outline in section 2.4, we expect different results from the two methods, with vials having several biases in terms of the materials (and quantities) they accumulate. The filter method was used to calculate C deposition rates as it is considered a 'passive' technique (see section 2.4), and is less likely to overestimate C deposition on a natural saltmarsh surface.

REFEREE COMMENT: As one of the goals of your study is to compare both the filter and vial method, please provide a more in-depth discussion of the effect of the order of magnitude difference between the results from both methods on the calculations you make and the conclusion you draw based on this data.

RESPONSE: As outlined in responses above, it is not our intent that the manuscript undertakes a formal comparison of different methods. We have also outlined how we intend to clarify this. We believe that sufficient discussion of the differences between filter and vial results has been made in the manuscript (particularly in sections 3.2 and 4.2.2). Use of vial-derived deposition rates for calculating C deposition would represent a substantial overestimate of actual C deposition, as it is an 'active' sedimentation method. As there is no rationale or intent to use the vial method to calculate C deposition rates, we do not see a reason for extending the discussion of the two methods here.

REFEREE COMMENT: P14 L26: By using the title 'Decomposition of organic matter...' you suggest that you have effectively measured OM decomposition, which is not the case. Please change the title so that this is more clear. E.g. 'Chemical structure of deposits varies among...'

RESPONSE: Agreed. We intend to change this title as suggested.

REFEREE COMMENT: P14 L28-31: Please reformulate this sentence: by saying '... these analyses have revealed insights in to fate of aboveground OM and the likelihood of their contribution to...' you suggest that you have done measurements that directly allow you to say something about the different contributions of OM in these different vegetation assemblages to long-term C sequestration. This is however not the case, as you use chemical measurements to make suggestions about these processes.

RESPONSE: We propose to delete this sentence, in light of the referee's comment.

REFEREE COMMENT: P15 L8: Based on which data do you calculate the 'retention of plant-derived C'? Please explain.

RESPONSE: The paragraphs preceding this statement, discuss the data upon which we come to this conclusion. That is:

"Importantly, cellulose also appears to be a factor in the separation of residues from the three different saltmarsh assemblages along PC2 (Fig. 4c), suggesting higher content in the two *Juncus* samples, followed by *Sporobolus* and then *Sarcocornia* samples. This finding was confirmed by 13C NMR data, which showed greater proportions of plant compounds (carbohydrates more broadly, as well as lignin) were retained within the *Juncus* litter and residue relative to the other species (Table 2). In contrast, the higher proportions of alkyl–C and amide/carboxyl–C within *Sarcocornia* and *Sporobolus* residues were indicative of higher protein and lipid contents, consistent with bacterial biomass and marine algae signatures (Dickens et al., 2006). However, they may also be partly explained by the selective retention of resistant plant waxes, such as suberin and cutan."

REFEREE COMMENT: P15 L17: 'The selective sorption of N by a plant…': how does this explain that Juncus litter is enriched in N compared to the original biomass?

RESPONSE: The point here is that the *Juncus* litter is depleted in N compared to the original biomass. This then gets reflected in a higher C:N ratio in the litter, relative to the live biomass

REFEREE COMMENT: P15 L23: How does table 2 show that the bacterial biomass increases for Sacocornia and Sporobolus?

RESPONSE: Table 2 does not show this directly. Instead this increase in bacterial derived C is inferred earlier in section 4.3.2:

"In contrast, the higher proportions of alkyl–C and amide/carboxyl–C within *Sarcocornia* and *Sporobolus* residues were indicative of higher protein and lipid contents, consistent with bacterial biomass and marine algae signatures (Dickens et al., 2006)."

It is for this reason that we refer to the 'increases suggested for *Sarcocornia* and *Sporobolus* assemblages'. We do note, however, that L23 should include reference to both bacterial biomass and marine algae, and another citation of Dickens et al. 2006.

REFEREE COMMENT: P15 L24-27: This seems highly speculative and you don't use any data or references to prove this: I suggest you remove this.

RESPONSE: Agreed. We intend to remove this sentence.

REFEREE COMMENT: P15 L30-31: you only measured C deposition on a very short timescale (averaged over 2 months), so I would refrain from any suggestions or conclusion of your observations for long-term C sequestration.

RESPONSE: We intend to modify this sentence to reflect the short-term nature of our measure, and be more circumspect in its suggestion:

"…highlight short-term processes which may contribute to the high capacity of *Juncus* to accumulate C stocks…"

Technical corrections

RESPONSE: Each of the technical corrections below will be incorporated in a revised manuscript. We thank the referee for taking the time to provide a comprehensive list of technical corrections.

P1 L15: remove 'surface'

P1 L21: Replace 'Accretion was…' by 'Accretions rates were…'

P1 L23: change '(6d)' to '(6 days)'

P1 L28: change 'mid infrared' to 'mid-infrared' (also in the rest of the ms)

P2 L5: change 'broad' to 'general'

P2 L8: change 'exceptional productivity' to 'exceptionally high productivity'

P2 L12: Change 'Surface elevation and sedimentation dynamics are central…' to 'Sedimentation dynamics partially determine the survival of coastal wetlands under rising…'

P2 L14-16: This is a strange sentence: first you define minerogenic as 'dominated by mineral inputs', by which you imply that there is also other (organic) material present. Next you say that most saltmarsh sediments contain both organic and mineral fractions, repeating what you first said. You can simply only say that most saltmarsh sediments are a mixture of organic and mineral materials, to avoid confusion.

P2 L18: change 'sediment' to 'sediments'

P2 L19-20: change '…); as well as the tidal range of a site and position…' to '…), the tidal range of a site and the position…'

P2 L25: change 'Broadly' to 'Generally'

P2 L26: change 'helping to trap mineral sediments' into 'facilitating sediment tapping'

P2 L27-30: Change to: 'Findings of comparative studies of the effect of vegetation composition on sediment deposition rates, however, vary from no difference among different vegetation species () to substantial differences among…'

P2 L32: I would change this sentence to: 'Average global rates of carbon accumulation in saltmarshes are extremely high, relative to…'

P2 L33: state that SE is the standard error

P2 L39: change 'their' to 'its'

P3 L1: change 'soil pools' to 'soils'

P3 L9: You can change this sentence to 'Because methods vary…, a combination of …'

P3 L15: change 'presented' to 'presents'

P3 L15-16: I would reformulate this sentence and state that another aim of your study was to compare different methods that are used to measure sedimentation rates on tidal marshes (otherwise it is not clear to the reader whether or not you made the comparison).

P3 L24-25: put '(Fig. 1)' at the end of the sentence

P3 L 25-26: 'mangrove species Avicennia…'

P3 L27: 'the upslope limit of saltmarshes…'

P3 L28: 'but for the most part saltmarshes are bordered…'

P3 L29-30: '… with ranges in elevation and tidal extent.'

P3 L31: 'Salmarshes within this site comprise…'

P3 L31: '… communities. The lower and middle marsh is characterized by an association of … pathway). The upper marsh …'

P3 L36: 'Fifteen plots were selected on the basis…'

P4 L5: is this g dry weight per m-2? If so, mention this, also in the next sentence.

P4 L6: '… 350 g m-2). Moreover, there do not'

P4 L12-15: Move these sentence to the beginning of the study area section: they provide general information about SE Australian saltmarshes.

P5 L5: Change 'sedimentation traps' to 'sediment traps'

P5 L35: Change to '… the supernatant decanted and the vial was placed…'

P7 L14-16: Please explain the symbols more clearly: e.g. 'where δ13C denotes the isotopic signal of different sources of OC: Cresidue (…), CC4 (…) and CC3 (…).

P8 L16: please mention the units of '100 ± 32.73'

P8 L19: better to give the range in $R^2$ instead of saying '$R^2 > 0.35$); I wouldn't call these relationships significant as long as you didn't test them statistically.

P10 L 15-16: change to '… and deposition measured with short-term sediment traps…'

Section 4.1: use the re-calculated accretion rates (see my comments above)

P11 L14: change 'massive' to 'large'

P12 L7: 'the physical position'

P14 L6: This sentence is not correct: change to e.g. '… surface dynamic is critical to predict the survival…'

P14 L11: Please rephrase 'organogenic and minerogenic assemblages' to e.g. 'organogenic and minerogenic deposits'

P14 L28: Replace 'MIR' by 'MIR spectroscopy'

P15 L38: remove 'then'

Figure 1

- Heading: change '…location of nearest…' in '…location of the nearest …'

Figure 2

- Heading: is 'SE' the standard error? Is this the same as standard deviation? Please clarify.

- Change the axes so that the 0 marker of the y-axis is at the same height of the x-axis (since you don't plot negative accretion)

- You should make it more clear that what you show is the height of deposited sediments above the marker horizon. Now the reader can interpret it as accretion rates measured at different time periods. I would change the y axis label to something like 'Height of deposited sediments (mm)'

Figure 3

- Heading: write '6d' as '6 days'

- As you have standard deviations on this data the quality of the figure would improve if the differences between the different vegetation species are significantly different, e.g. with letters above the bars.

Figure 4

- The letters written within the symbols of A) are very difficult to read: place them next to the symbols

- Also the letters next to the symbols in A) are difficult to read: enlarge them and increase the space between the symbol and the letters

Figure S1

- Heading: replace 'scatterplots' with 'plots'; explain what 'AHD' is; put 'regression line' in plural; explain that DW (on the y-axis) means dry weight; explain what 'bulk material' is.

- Y-axis: change units to 'g DW m-2'

- Plot D should be January 'spring' instead of 'neap'?

Figure S2

- Heading: same remarks as for fig. S1

- Replace the y-axis label as for fig. S1

- Remove 'no linear fits' from the legend: this is already explained in the heading

- Plot D should be January 'spring' instead of 'neap'?

Tables

Table 1

- Heading: change 'Summary of sediment measure techniques…' to 'Summary of sedimentation measurement techniques'; Change 'C' to 'OC', since you measure only organic carbon

- Under Parameter, change 'Measure' into 'Measurement'

- Under 'Filter + isotopic analyses': clarify what 'sediment residue' is. This should be clear to the reader without reading the whole manuscript.

- Under 'Filter + MIR & 13C NMR': change 'Character of …' to 'Characteristics of …'

- In the 'Filter + elemental analysis' section: C deposition rate is expressed in 'yr-1' while you only measured for a short period in summer. This should be changed (see my previous comments)

- In the notes (a): change '%C' to '%OC', since you measured organic carbon

- For the filter method – 'Filter + isotopic analysis': it should be clear what 'sediment residue' is, please clarify in the heading.

Table 2

- Heading: change 'assemblage' to plural; change '… plant assemblages, plus other…' to 'plant assemblages and other potential sources'; change '… for each of biomass…' to '…for each of the biomass…'

- Explain what 'n/a' stands for in the heading

Table S1

- Place 'Number of tides exceeding mean plot elevation' above the names of the neap and spring events to increase readability

Table S2

- Are these values based on 1 measurement or are these average values from multiple replicates? If so, provide the standard deviation

RESPONSE: As noted above, each of the technical corrections will be incorporated in a revised manuscript. We thank the referee for taking the time to provide a comprehensive list of technical corrections.

**ADDITIONAL REFERENCES**

Fry, B. and Ewel, K. C.: Using stable isotopes in mangrove fisheries research--a review and outlook, Isotopes Environ Health Stud, 39, 191-196, 2003.

Hemminga, M. and Buth, G.: Decomposition in salt marsh ecosystems of the SW Netherlands: the effects of biotic and abiotic factors, Plant Ecology, 92, 73-83, 1991.

Hessen, D. O., Elser, J. J., Sterner, R. W., and Urabe, J.: Ecological stoichiometry: An elementary approach using basic principles, Limnology and Oceanography, 58, 2219-2236, 2013.

Jaschinski, S., Hansen, T., and Sommer, U.: Effects of acidification in multiple stable isotope analyses, Limnology and Oceanography: Methods, 6, 12-15, 2008.

Mueller, P., Jensen, K., and Megonigal, J. P.: Plants Mediate Soil Organic Matter Decomposition In Response To Sea Level Rise, Global Change Biology, 22, 404-414, 2016.

Rogers, K., Saintilan, N., Howe, A. J., and Rodríguez, J. F.: Sedimentation, elevation and marsh evolution in a southeastern Australian estuary during changing climatic conditions, Estuarine, Coastal and Shelf Science, 133, 172-181, 2013.

Saintilan, N., Rogers, K., Mazumder, D., and Woodroffe, C.: Allochthonous and autochthonous contributions to carbon accumulation and carbon store in southeastern Australian coastal wetlands, Estuarine, Coastal and Shelf Science, 128, 84-92, 2013.

Sterner, R. W. and Hessen, D. O.: Algal nutrient limitation and the nutrition of aquatic herbivores, Annual review of ecology and systematics, 25, 1-29, 1994.

Zieman, J., Macko, S., and Mills, A.: Role of seagrasses and mangroves in estuarine food webs: temporal and spatial changes in stable isotope composition and amino acid content during decomposition, Bulletin of Marine Science, 35, 380-392, 1984.